# Explaining the Inconsistency of Perturbation-Based Fidelity Metrics

## Abstract

Saliency maps are one of the most widely used post-hoc approaches for interpreting the behavior of Deep Learning models. Yet, assessing their fidelity is difficult in the absence of ground-truth explanations. To address this, numerous fidelity metrics have been introduced. Previous studies have shown that fidelity metrics can behave inconsistently under different perturbations, and a recent work has attempted to estimate the extent of this inconsistency. However, the underlying reasons behind these observations have not been systematically explained. In this work, we revisit this problem and analyze why such inconsistencies arise. We examine several representative fidelity metrics, apply them across diverse models and datasets, and compare their behavior under multiple perturbation types. To formalize this analysis, we introduce two conformity measures that test the assumptions implicit in existing metrics. Our results show that these assumptions often break down, explaining the observed inconsistencies and calling into question the reliability of current practices. We therefore recommend careful consideration of both metric choice and perturbation design when employing fidelity evaluations in eXplainable Artificial Intelligence (XAI).

## 1 Introduction

Deep learning (DL) models have shown significant improvement in accuracy as compared to traditional Machine Learning (ML) models. However, such improvements have come at the cost of decreased transparency. Hence, concerns about the transparency, fairness, privacy, and trustworthiness of AI applications arise due to the black-box nature of DL models in high stakes domains like health care, insurance, and law enforcement Rudin (2019), Jacovi et al. (2021), Arrieta et al. (2020). These concerns have led to skepticism about adopting the latest Artificial Intelligence (AI) models in various sectors Cubric (2020), Cam et al. (2019), Güngör (2020). Therefore, research has been dedicated to explaining the decisions of DL models under the umbrella of XAI Arrieta et al. (2020), Selvaraju et al. (2017), Chattopadhay et al. (2018), Zhou et al. (2016), Ramaswamy et al. (2020), Ribeiro et al. (2016),Broniatowski et al. (2021), Lundberg & Lee (2017).

Saliency maps, such as Class Activation Maps (CAM), are widely used to explain the predictions of Deep Learning (DL) models by highlighting the image regions that are most important for the model's decision Selvaraju et al. (2017), Chattopadhay et al. (2018). However, disagreements are commonly observed among saliency maps generated using different methods for the same model and the same image, leading to confusion. One can choose the best saliency map with the highest fidelity when compared to the ground truth. However, the absence of actual ground-truth[1] has led to the development of fidelity metrics like, "Area Over the Perturbation Curve" ($AOPC$) (Samek et al., 2016), Average Drop (AD%), Increase in Confidence (IC%) and Win (W%) (Chattopadhay et al., 2018), Wang et al. (2020) and "faithfulness" metric (Alvarez Melis & Jaakkola, 2018).

These fidelity metrics, however, suffer from inconsistencies and thus make them unreliable (Tomsett et al., 2020). Fidelity metrics such as $AOPC$, AD%, IC% and W% and $faithfulness$ rely on computing pixel importance rank (PIR) for measuring the fidelity of saliency maps Bora et al. (2026).

---

[1] Human annotation represents the regions of an image from a human perspective (e.g., edges in images), but they do not have any relation to the patterns the DL model is considering for decision making. Thus, human-annotated saliency maps may misrepresent the model's true decision-making process, making them unreliable for evaluating the fidelity of the maps.

PIR is calculated by perturbing the pixels (one by one or cumulatively) and noting the change in the output probability[2]. A greater change in output probability denotes greater importance for a perturbed pixel. The computed PIR from an image serves as a proxy for ground truth, enabling the estimation of the fidelity score for saliency maps (Alvarez Melis & Jaakkola, 2018). This approach is based on the assumption that the change in output probability follows a consistent pattern across different perturbations, with the output probability varying in proportion to the importance of the perturbed pixel. If this assumption is not fulfilled, the fidelity metrics' scores would vary for different perturbations, leading to inconsistency as reported by Tomsett et al.(Tomsett et al., 2020). Further, Tomsett et al.(Tomsett et al., 2020) observed this inconsistency by analyzing the prediction probabilities by perturbing pixels with 0 and a random value. While highlighting the inconsistency in fidelity metrics, Tomsett et al. (Tomsett et al., 2020) emphasize that developers of such metrics should guide practitioners to examine the sources of variance in metric scores and to understand how this variability influences the choice of saliency methods for a given model.

## 1.1 RELATED WORKS

Prior studies have reported that perturbation-based fidelity metrics can be statistically unreliable (e.g., Tomsett et al. (2020)). Additionally, FRIES (Bora et al. (2026)) took a complementary approach and introduced an estimation framework that predicts how inconsistent a metric will be for a given model–dataset–perturbation setting, using features derived from output probability variations under perturbations and training a supervised model using it. In contrast, this paper addresses a different question: we seek to explain why these inconsistencies arise in the first place and thereby provide a lightweight alternative to FRIES without requiring to train a supervised model. We formalize the assumptions implicitly required by widely used metrics and show, analytically and empirically, the specific conditions under which those assumptions fail.

This paper uses the same foundational primitives that underlie the FRIES framework but with a different role. In FRIES, those primitives served as features for predictive inconsistency estimation. However, in this paper they are the central theoretical objects and we use them to explain failure conditions for fidelity metrics and to define diagnostic conformity measures that directly test the validity of the metrics' assumptions without training an estimator model.

## 1.2 OUR CONTRIBUTIONS

We first theoretically establish the scenarios under which such assumptions are violated. We then provide two conformity measures that quantify the extent of variances affecting the fidelity metrics. Both the conformity measures are used to demonstrate the inconsistency of fidelity metrics by using several perturbations, models and datasets in both normal and adversarial setting. Going beyond the works of Tomsett et al.(Tomsett et al., 2020) and to generalize our findings, we study the variances in a comprehensive manner using nine different perturbations that include two inpainting-based perturbations (Telea (Telea, 2004) and Navier Strokes (Bertalmio et al., 2001)), Gaussian Blur (three different widths of the Gaussian Kernel) and setting a random value, min, max and mean of the image pixel values as perturbation values. Further, we show empirically that our conformity measures can be used in pixel-wise and segment-wise perturbation schemes before using fidelity metrics.

Our main contributions to this paper are:

- We present an approach to explain the inconsistency of fidelity metrics. We show that before using fidelity metrics, the variances of DL models w.r.t. to the perturbation type must be studied.
- Complementing previous works that have observed inconsistencies in fidelity metrics Tomsett et al. (2020), and proposed methods to estimate the inconsistency using supervised learning Bora et al. (2026), this paper explains inconsistency by (i) formalizing the assumptions underpinning common fidelity metrics, (ii) proving where these assumptions break under realistic perturbations, and (iii) introducing DROP and PSim as lightweight conformity measures (without requiring to train a supervised model like FRIES) to assess assumption validity before a fidelity metric is applied.

---

[2]All the reported observations in this paper are based on the prediction probability of the top prediction class. We will refer to this as the output probability from hereon.

- The conformity measures proposed in this work are further used to empirically analyse three widely used DL models and two adversarially trained DL models on three datasets using nine perturbation types, and two perturbation schemes (pixel-wise and segment-wise) for all models.

## 2 PROPOSED APPROACH

The fidelity metrics are based on the PIR which assume the drop in output prediction probability of a DL model to be proportional to the relevance of the perturbed pixel (i.e., more important the pixel, larger the drop in output probability). The pattern of change (i.e. the proportionate change in output probability as per the relevance of the perturbed pixel) should ideally hold true for all types of perturbations as long as the image semantics is preserved under the notion of local neighborhood Bora et al. (2026). This is based on two aspects:

[P1] There is a drop in the output probability when a pixel is perturbed;

[P2] The magnitude of drop in output probability is proportional to the relevance of the pixel.

Dissecting these two aspects, we first present the theoretical background on the violation such aspects in fidelity metrics and then present the proposed conformity measures in Section 2.2 and Section 2.3 to explain the inconsistencies.

### 2.1 THEORETICAL FRAMEWORK

Let $\mathfrak{R}$ be the ranks of pixel as per importance obtained from a saliency map on an unperuturbed image. $\mathfrak{R}$ can be expressed as follows:

$$\mathfrak{R} = \{a_1, a_2, a_3, a_4, \ldots a_i\} \tag{1}$$

where, $\mathfrak{R}$ is the ranked list of pixel importance by any saliency method. $a_1 \to a_i$ are pixels sorted in the order of their importance i.e. a greater $i$ denotes greater importance.

The assumption on the expected change in output probability by perturbing a pixel can be summarized as:

$$p_0 > p_i^\phi \quad \forall \quad i, \phi \tag{2}$$

where, $p$ is the prediction probability of a classification model which takes an image $I$ as input and returns the probability of the top class. $p_0$ is the probability of the top class as predicted for the original i.e. unperturbed image. $p_i^\phi$ is the prediction probability on an image obtained by perturbing only the $i^{th}$ pixel of an image $I$ with a perturbation type $\phi$.

Further, the change in output probabilities of perturbing two pixels $i$ and $j$, where $j$ is more important than $i$, can be summarized given as:

$$\delta p_i^\phi < \delta p_j^\phi \quad \forall \quad i < j \tag{3}$$

Where, $\delta p_i^\phi = p_0 - p_i^\phi$

Utilizing Equation (1) and Equation (3) we can generate the ranked list of probability differences, denoted as $\mathfrak{R}(\phi)$, for an image perturbed by each pixel and for all $i$ pixels with increasing order of ranks:

$$\mathfrak{R}(\phi) = \{\delta p_1^\phi, \delta p_2^\phi \ldots \delta p_i^\phi\} \tag{4}$$
$$pixels = \{1, 2, \ldots i\} \text{ and for a given perturbation } \phi$$

The probability changes obtained from Equation (4) can be sorted to get an ordered list of pixels. This set of ordered pixels, denoted by $R_\sigma$, represents the importance ranks of the pixels corresponding to $\sigma$. For a perturbation based technique to be applicable in fidelity metrics, the pixel importance

ranks should ideally be invariant to different sets of hyper-parameters. This invariance to different sets of hyper-parameters is defined as below:

$$rbo(\Re(\phi), \Re(\psi)) \approx 1 \quad \forall \quad \text{for two perturbations} \quad \phi, \psi \tag{5}$$

Where, $rbo$ is Rank Biased Overlap Webber et al. (2010) in our experiments, but it can be any function that calculates the similarity between two rank lists. Further, without the loss of generality we can say that Equation (5) should hold true for any set of pixels obtained from a saliency map.

Any perturbation based fidelity metric should conform to Point [P1] according to Equation (2) and should conform to Point [P2] according to Equation (5). To quantify the conformance, we introduce two new conformity scores which we refer to as $DROP$ (corresponds to Point [P1]) and $PSim$ (corresponds to Point [P2]) as discussed further.

## 2.2 Drop in Prediction Probability (DROP)

The drop in Prediction Probability ($DROP$) metric measures the average number of drops in the output probability when a pixel is perturbed for an image and a given model $\mathcal{M}$ across all perturbation types $\mathcal{N}$. Thus, if $p_0$ represents output probability from a model $\mathcal{M}$ on unperturbed image and $p_s^\phi$ represents the output probability on a perturbed image for a perturbation type $\phi$ on a chosen pixel $s$ in a set of all pixels $\mathcal{S}$ or a chosen segment of all available segments, $DROP_\mathcal{M}$ for a given model can be computed as:

$$DROP_\mathcal{M} = \frac{1}{|\mathcal{N}|} \sum_{\phi \in \mathcal{N}} \frac{\sum_{s \in \mathcal{S}} \left[ p_0 >= p_s^\phi \right]}{|\mathcal{S}|} \tag{6}$$

Where, [] denotes an indicator function with binary decision. For a complete dataset of $K$ images and a given model $M$, the $DROP$ scores from Equation (6) are averaged across all images in a dataset $D$. The ideal value of $DROP$ should be 1 with higher value representing higher conformity to Point [P1].

## 2.3 Pixel Rank Similarity (PSim)

We define the metric $PSim$ to measure the average similarity of PIRs across all perturbations for an image. For any two given perturbations ($\phi$ and $\psi$ from a set of perturbations $\mathcal{N}$) on an image, and corresponding PIRs obtained $\Re(\phi), \Re(\psi)$ respectively for a given image, it is expected to have same ranks for a given model $M$ if the model is consistent. Thus, the average similarity between the ranks across all perturbations can be computed as:

$$PSim_\mathcal{M} = \frac{\sum_{\phi \in N} \sum_{\psi \in N, \phi \neq \psi} rbo(\Re(\phi), \Re(\psi))}{\frac{|\mathcal{N}| \times (|\mathcal{N}|-1)}{2}} \tag{7}$$

Thus, for any perturbation based fidelity metric to be consistent, $PSim$ should have an ideal value of 1. However, higher values i.e., closer to 1 suggest higher conformance to Point [P2].

## 3 Implementation Details

### 3.1 Approach Overview

Figure 1 shows our implementation where we obtain the prediction probabilities for a given model on unperturbed and a set of perturbed images. The prediction probabilities are used to evaluate the conformance using Drop in Prediction Probability (DROP) for Point [P1] and Pixel Rank Similarity (PSim) for Point [P2]. The approach for measuring the conformity scores is further described in Algorithm 1. While Algorithm 1 computes the conformity scores for the pixel-wise perturbation scheme, the same can be applied to the segment-wise perturbation scheme without the loss of generality.

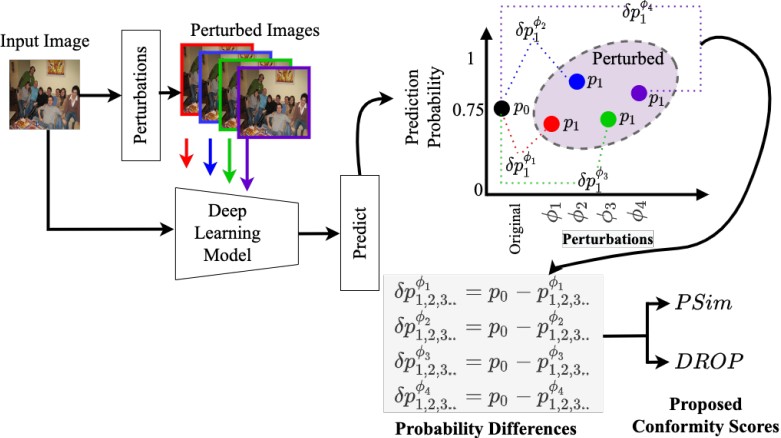

Figure 1: Proposed approach for estimating conformity scores of the deep learning models using the prediction probabilities on perturbed images.

We first determine the prediction probability of a given model $M$ on an unperturbed image (i.e., $p_0$) and then perturb the selected pixels one by one for a given perturbation $\phi_1$ to obtain $p_1, p_2, p_3 \ldots$ to determine the $\delta p_1, \delta p_2, \delta p_3 \ldots$ for the perturbation $\phi_1$. The same perturbation scheme can be extended to segments without any change. $DROP$ and $PSim$ are then calculated for each image and for the whole dataset as described in Equation (6) and Equation (7) respectively.

---

**Algorithm 1** Algorithm for calculating $DROP$ and $PSim$

---

$p_0 \leftarrow model.predict(I)$            ▷ Unperturbed image $I$
$\{i\} \leftarrow S$            ▷ $i$ pixel in $S$ pixels
$\phi \leftarrow \{\Phi\}$            ▷ set of all perturbation types
$\mathcal{L} \leftarrow []$
$\mathcal{L}$            ▷ List of pixel importance ranks from all perturbation types
$\delta \mathcal{P} \leftarrow []$            ▷ $\delta \mathcal{P}$ is the $DROP$ score
**for all** $\phi$ **do**
     $\delta P \leftarrow []$
     **for all** $i\ in\ \{S\}$ **do**
         $I_i^\phi \leftarrow perturb\_image(I_i, \phi)$            ▷ for $i^{th}$ pixel in image $I$
         $p_i^\phi \leftarrow model.predict(I_i^\phi)$
         $\delta p_i^\phi = p_0 - p_i^\phi$
         $\delta P.append(\delta p_i^\phi)$
     **end for**
     $\delta \mathcal{P}.append(|\{\delta P \geq 0\}|)$            ▷ Append count of $\delta P \geq 0$
     $l \leftarrow argsort(\delta P)$
     $\mathcal{L}.append(l)$
**end for**
$rbo\_score \leftarrow pairwise\_rbo(\mathcal{L})$
**return** $\mu(\delta \mathcal{P})$, $\mu(rbo\_score)$            ▷ DROP (Equation (6)) and $PSim$ (Equation (7)) scores

---

## 4 EXPERIMENTAL SETUP

We use three pre-trained, and two adversarially trained image classification models, and three well-known datasets in our experiments. We conduct our analysis on InceptionV3 Szegedy et al. (2016), Xception Chollet (2017), and ResNet50 He et al. (2016) initialized with ImageNet weights. For, adversarial models we used the weights of adversarially trained ResNet50 architecture viz., ImageNet L2-norm (ResNet50) with $\epsilon = 3$ and ImageNet Linf-norm (ResNet50) with $\epsilon = 8/255$ ( refer Engstrom et al. (2019) for details). Imagenette from tensorflow.org et.al., Oxford-IIIT Pet Dataset

Parkhi et al. (2012) and PASCAL VOC 2007 Everingham et al. are used to conduct our experiments. The Imagenette dataset is a subset of the Imagenet et.al. dataset with ten easily classified classes. We used the validation part of this dataset for our experiments, which has around 3925 images. The Oxford-IIIT Pet Dataset Parkhi et al. (2012) and PASCAL VOC 2007 Everingham et al. datasets did not have train and test splits. Hence, we considered all the images for these two datasets, i.e., 7390 of the Oxford-IIIT Pet dataset and 4952 of the PASCAL VOC 2007 dataset. For each model, $predict$ was called for $(3925 + 7930 + 4952)\ images \times 50\ pixels \times\ 9\ perturbatiotypes \times 2\ perturbationschemes$ values, approximately, 15 million times, and in total, predict was called approximately 75 million times. Further, our goal was not to be exhaustive with different datasets and models but to understand the impact of perturbations to evaluate the fidelity of saliency maps from the perspective of PIR. Our code was written in Python 3.10 and Tensorflow 2.9 and for computing we leveraged A100 GPUs.

## 4.1 PERTURBATION DETAILS

We considered nine different perturbation types i.e., two inpainting based perturbations for all our experiments. Specifically, we used Telea Telea (2004) and Navier Strokes Bertalmio et al. (2001)), Gaussian Blur (three different widths of the Gaussian Kernel) and setting a random value, min, max and mean of the image pixel values as pixel values (as used by Tomsett et al. (2020), and Bora et al. (2026)). The perturbations are represented as 'IT' (Telea inpainting),'IN' (Navier Strokes inpainting), 'FR' (setting pixel value randomly), 'U0' (image min), 'U1' (image max), 'U0.5' (image mean), 'G3' (Gaussian blur with kernel widths of 0.3), 'G9' (Gaussian blur with kernel widths of 0.9) and 'G1.5' (Gaussian blur with kernel widths of 1.5). Further, we perturb the pixels/segments using two perturbation schemes viz., pixel-wise and segment-wise. We used Quickshift Vedaldi & Soatto (2008) segmentation algorithm to compute the segments for segment-wise perturbations. We use the property that a subset of a ranked order list maintains the original ranking and select 50 random pixels (refer to proof in Section S2). The same argument was extended to segments in our analysis.

## 5 RESULTS AND DISCUSSION

### 5.1 DROP AND PSIM SCORES FOR ALL PERTURBATIONS

Table 1 shows the $DROP$ and $PSim$ values for different models over different datasets for pixel-wise and segment-wise perturbation scheme. The chosen models, i.e., Inception V3, Xception, and ResNet50 pretrained with Imagenet weights. As seen in Table 1, it can be observed that the $DROP$ values were around 0.5 to 0.6 for all models across datasets. This indicates that only for 50 % to 60% of the pixels, the probability dropped after perturbation but for other pixels the output probability increases. This invalidates Point [P1] of the assumption in Section 2. Further, Table 1 shows the $PSim$ values for all the models over all datasets. As seen from the table, the $PSim$ values are small, but as per Equation (7), they should have been $\approx 1$. This invalidates Point [P2] of the assumption in Section 2. Further, this observation is consistent for all three models and across all datasets for segment-wise perturbation scheme as seen in Table 1. Thus, for different perturbations, the mentioned models will not conform to the assumptions made by the perturbation based fidelity metrics.

Further, we show the $DROP$ and $PSim$ scores for the adversarially trained ResNet50 models for both perturbation schemes in Table 2. Both $DROP$ and $PSim$ scores are much lower than 1 in all cases, and hence, adversarial training does not necessarily result in consistency of fidelity metrics. Due to the unavailability of adversarially trained models for Inception_V3 and Xception architectures, we had to limit our experiments to ResNet50 architecture. Hence, we refrain from making conclusive remarks regarding the consistency of fidelity metrics with respect to adversarially trained models.

### 5.2 DROP FOR INDIVIDUAL PERTURBATIONS

We present the distribution of $DROP$ scores for Inception V3, Resnet50, and Xception models in the Imagenette dataset for pixel-wise perturbation scheme in Figure S2. For all perturbations, except

Table 1: $DROP$ and $PSim$ scores across all datasets, models, perturbations for pixel-wise perturbation scheme and segment-wise perturbation scheme. The results are shown as Mean $\pm$ Standard Deviation. Ideal value $DROP$ and $PSim$ should be 1 and higher the better.

| Dataset | | Inception | Xception | ResNet |
|---|---|---|---|---|
| | | Pixel-wise perturbation | | |
| Imagenette | $DROP$ | 0.504±0.131 | 0.514±0.134 | 0.643±0.153 |
| | $PSim$ | 0.432±0.181 | 0.431±0.185 | 0.570±0.298 |
| Oxford Pets | $DROP$ | 0.507±0.130 | 0.504±0.138 | 0.636±0.132 |
| | $PSim$ | 0.428±0.183 | 0.430±0.186 | 0.582±0.289 |
| VOC2007 | $DROP$ | 0.511±0.115 | 0.550±0.180 | 0.512±0.132 |
| | $PSim$ | 0.643±0.130 | 0.433±0.189 | 0.573±0.301 |
| | | Segment-wise perturbation | | |
| Imagenette | $DROP$ | 0.515±0.135 | 0.518±0.126 | 0.553±0.111 |
| | $PSim$ | 0.310±0.181 | 0.269±0.142 | 0.329±0.179 |
| Oxford Pets | $DROP$ | 0.507±0.120 | 0.516±0.095 | 0.546±0.107 |
| | $PSim$ | 0.255±0.129 | 0.307±0.179 | 0.309±0.181 |
| VOC2007 | $DROP$ | 0.542±0.102 | 0.517±0.091 | 0.529±0.100 |
| | $PSim$ | 0.267±0.166 | 0.294±0.179 | 0.299±0.182 |

Table 2: $DROP$ and $PSim$ scores for adversarially trained ResNet50 models (Linf-norm and L2-norm) for pixel-wise and segment-wise perturbation schemes. The results are shown as Mean $\pm$ Standard Deviation. (*Higher scores are better with ideal being closer to 1)

| | Pixel-wise Perturbation | | | |
|---|---|---|---|---|
| Dataset | L2-norm($DROP$) | Linf-norm($DROP$) | L2-norm($PSim$) | Linf-norm($PSim$) |
| Imagenette | 0.555±0.374 | 0.555±0.357 | 0.237±0.140 | 0.209±0.097 |
| Oxford Pets | 0.580±0.369 | 0.567±0.369 | 0.217±0.133 | 0.186±0.116 |
| VOC2007 | 0.528±0.383 | 0.546±0.371 | 0.243±0.124 | 0.181±0.106 |
| | Segment-wise Perturbation | | | |
| Dataset | L2-norm($DROP$) | Linf-norm($DROP$) | L2-norm($PSim$) | Linf-norm($PSim$) |
| Imagenette | 0.574±0.238 | 0.526±0.220 | 0.321±0.173 | 0.301±0.146 |
| Oxford Pets | 0.541±0.218 | 0.567±0.213 | 0.318±0.165 | 0.326±0.182 |
| VOC2007 | 0.557±0.186 | 0.517±0.181 | 0.292±0.148 | 0.289±0.155 |

the variants of Gaussian Blur, the $DROP$ scores have the highest density at around 0.5. However, the variations of the Gaussian Blur for the ResNet50 model seem to be closer to 1. This pattern is similar for other datasets (please refer to Section S3 in supplementary for exhaustive plots). Further, we estimated the probability of the $DROP$ scores to be closer to 1 (i.e., above the cut-offs of 0.80, 0.85, 0.90, and 0.95) by using Kernel Density Estimation (KDE), with Scott's rule Scott (2015) for bandwidth calculation, owing to its non-parametric nature. The estimated probabilities for $DROP$ scores to be above the cutoffs across all datasets, models, and perturbation types and schemes were low, but the variants of Gaussian Blur showed relatively higher probabilities than other perturbations (refer Figure S25 in supplementary for details). We observe a similar trend for the segment-wise perturbation scheme (refer Figure S26 in supplementary). This demonstrates empirically that fidelity metrics have low conformity to Point [P1] and our KDE based cutoff estimations further evidence for our claim.

### 5.3 PSim for Individual Pairs of Perturbations

The pairwise $PSim$ scores for all perturbation pairs corresponding to the Inception V3 model on the Imagenette dataset are shown for the pixel-wise perturbation scheme in Figure 3. Most of the perturbation pairs have low $PSim$ scores, but for the three pairs of Gaussian Blur (i.e., G3_G9, G3_G15, and G9_G15) and the pair for inpainting (IT vs. IN), the $PSim$ scores are relatively higher. We show the $PSim$ scores for all perturbation pairs on all dataset:model combinations in Section S4 of supplementary. Additionally, we estimated the probability of $PSim$ scores to be above the cutoff threshold of 0.80, 0.85, 0.90, and 0.95 using KDE (like Section 5.2). The estimated probabilities for $PSim$ scores to be higher than the cutoff thresholds were low in all situations (refer to Section S6 in supplementary for details). It was observed that in none of the scenarios, $PSim$ score is $\approx 1$,

indicating low conformity to Point [P1]. Hence, the ranks of the pixels/segments (as mentioned in Section 2.1) would vary for different perturbation types and lead to inconsistency in fidelity metrics.

From the low probabilities observed in Section 5.2, and Section 5.3, it can be established that fidelity metrics have low conformity to Point [P1] and Point [P2] and hence are not consistent across a wide variety of perturbations. As such, it is imperative to specify the perturbation type to be used when reporting the fidelity scores from these fidelity metrics. The perturbation type can be determined using domain-related theoretical reasoning and/or empirically. Further, we also observed that, out of the perturbation types considered, Gaussian Blur was relatively consistent compared to other perturbation types as it had higher scores for both conformity measures.

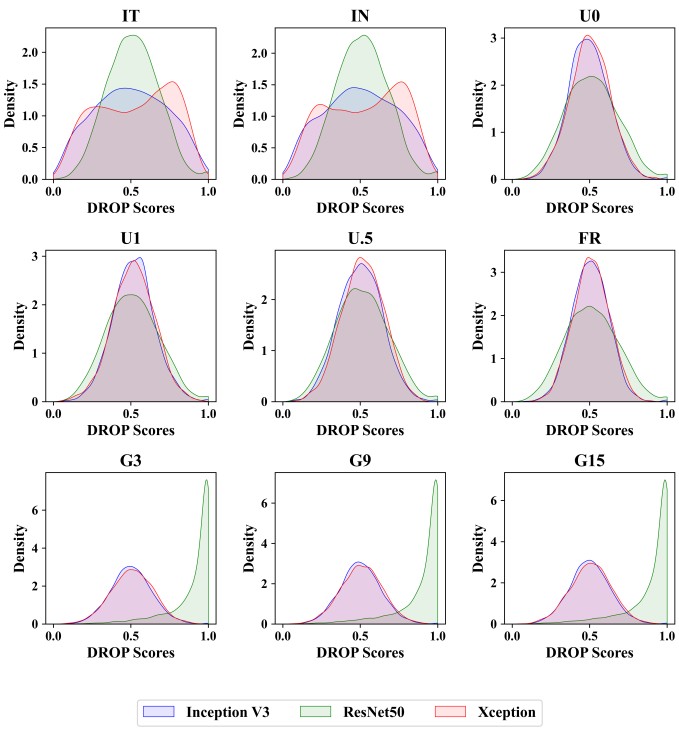

Figure 2: Distribution of $DROP$ scores across all models, perturbation types using pixel-wise perturbation scheme for Imagenette Dataset

## 6 CONCLUSION AND FUTURE WORK

The prediction probabilities of DL models vary significantly for the same image and model across the perturbations we considered. This results in a violation of the two assumptions of fidelity metrics: a drop in the output probability upon perturbing an image and no variance in PIR for different perturbations. Hence, fidelity metrics that rely on the mentioned assumptions become unreliable. Prior work has primarily framed unreliability at the metric level and attempted to estimate inconsistency, but our results show that it is fundamentally a model–perturbation interaction phenomenon. Beyond computing DROP and PSim, we used KDE-based tail-probability estimates to quantify how often these metrics approach 1; the consistently low probabilities provide robust evidence that both metrics fall well below the ideal value of 1. We therefore recommend using our proposed metrics as a precondition before any saliency-fidelity analysis, and consistently reporting the exact perturbation type and parameters alongside fidelity scores. Additionally, for fidelity metrics to be meaningful, the perturbation must be theoretically justified rather than setting pixels to arbitrary values, such as 0 or 1. Among the perturbations we tested, Gaussian blur exhibited comparatively consistent behavior. Future work should account for the violations discussed in this work while devising fidelity metrics and extend this analysis to adversarially trained models and additional architectures using our conformity measures.

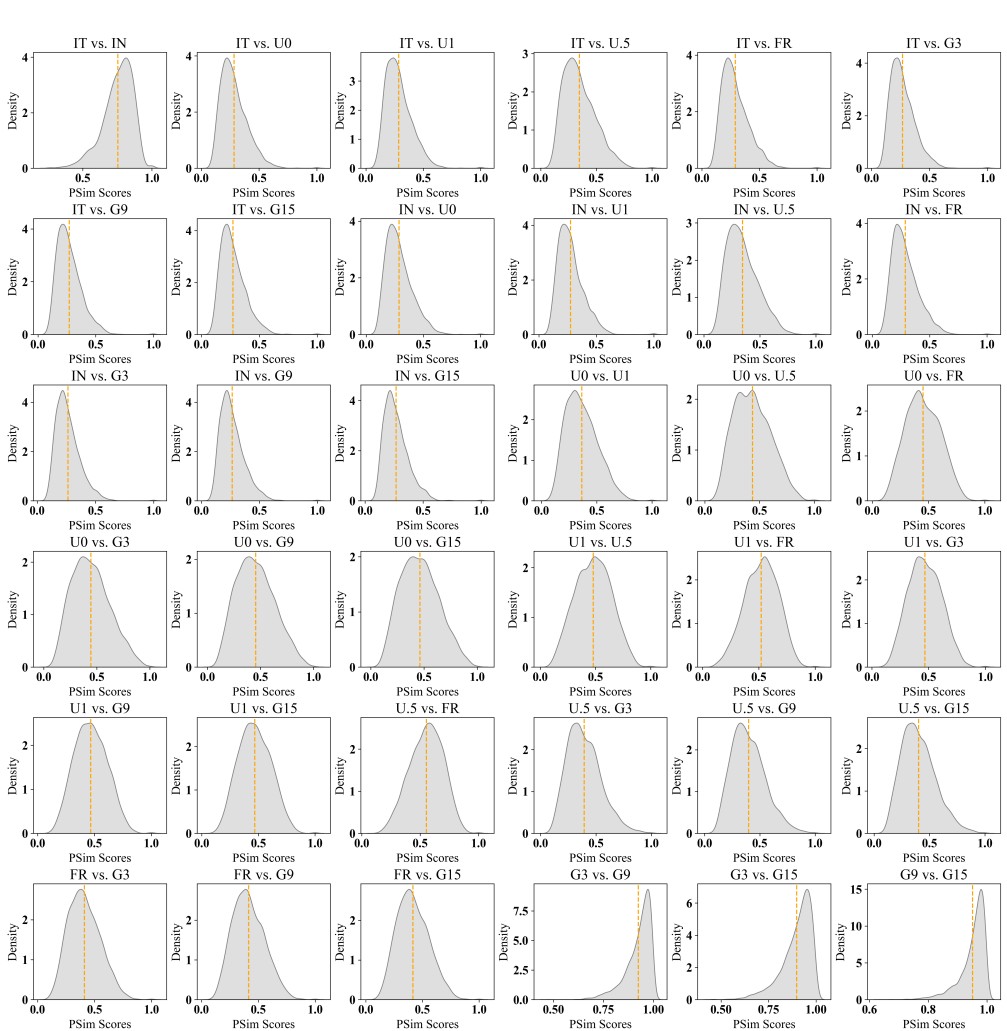

Figure 3: Distribution of pairwise $PSim$ scores for all perturbation types for Inception V3 model using pixel-wise perturbation scheme on Imagenette Dataset

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

## A  APPENDIX: EXAMINING WHY PERTURBATION-BASED FIDELITY METRICS ARE INCONSISTENT

## S1  CAM DISAGREEMENTS

The illustration in Figure S1 presents saliency maps on randomly sampled images from the CIFAR-10, Imagenette, Oxford-IIIT Pets and PASCAL VOC 2007 datasets for pretrained ResNet50 model (imagenet weights) using AblationCAM Ramaswamy et al. (2020), GradCAM++ Chattopadhay et al. (2018) and GradCAM Selvaraju et al. (2017). It can be noted from Figure S1, the saliency maps generated using AblationCAM and GradCAM++ show a high degree of agreement, highlighting the importance of the body, neck, and head of the horse for an image from Cifar-10 dataset (1st row). However, the saliency map generated using GradCAM completely misses highlighting the head of the horse. In the 2nd row it can be observed that AblationCAM and GradCAM++ highlight not only the head of the fish but also other areas in the background as compared to GradCAM. Similarly, the saliency maps generated for the Oxford-IIIT Pets dataset image (3rd row) and PASCAL VOC 2007 image (4th row) show high inconsistency.

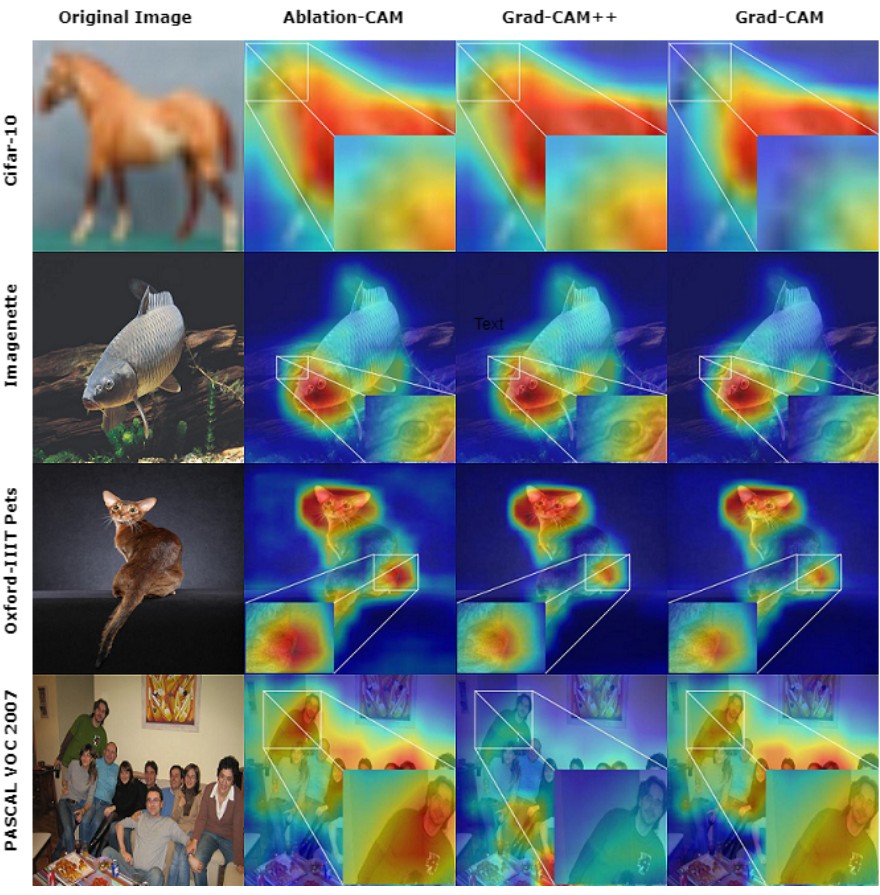

Figure S1: Disagreement between saliency maps generated using Ablation-CAM, Grad-CAM++ and Grad-CAM for ResNet50 model with imagenet weights. Each row represents a randomly chosen image from CIFAR-10, Imagenette, Oxford-IIIT Pets and PASCAL VOC 2007 datasets and their corresponding saliency maps

## S2 PIXEL/SIGMENT SELECTION AND RANKING

Selection of pixels/segments for our analysis is another critical aspect for our analysis. As the size of the input images are typically $299 \times 299$, $224 \times 224$ or $600 \times 600$ pixels for models, it is computationally expensive to conduct an analysis on all pixels. We therefore conduct our analysis on a subset of pixels which were randomly selected (based on Tomsett et al. (2020) and Bora et al. (2026)). Our approach to randomly select the pixels can be further justified from a theoretical perspective as explained below.

Let $Q$ be a set of pixels such that $|Q| > 1$. We can define a hypothetical function $\psi(Q)$ that measures the importance of $Q$ for the decision-making process of the model as:

$$\psi : Q \to \{1, 2, \ldots, |Q|\} \subseteq \mathbb{R}$$

where $\mathbb{R}$ is the set of all real numbers and a greater value of $\psi(Q)$ indicates greater importance.

We can define an image $\mathbb{A}$ as an ordered set of pixels sorted according to their importance using function $\psi$.

$$\mathbb{A} = \{a_1^u, a_2^v, a_3^w, \ldots a_i^z\} \tag{8}$$

where, $R_0$ is the ordered set of pixels. $1 \to i$ are importance for the pixel index/ids $u \to z$ generates by $\psi$ i.e. $\psi(a^u) = 1$, $\psi(a^v) = 2 \ldots \psi(a^z) = i$ etc, where a greater value of $\psi(Q)$ indicates greater importance of the pixel set $Q$ in the image.

Let us assume that $\mathbb{B}$ is a randomly selected subset of pixels. Thus $\mathbb{B}$ can be defined as below:

$$\mathbb{B} = \{a_1^x, a_2^y, a_3^z, \ldots a_j^n\} \subseteq \mathbb{A} \quad \text{s.t.}$$
$$a^e \neq a^f \quad \text{for} \quad e \neq f \tag{9}$$

where $e$ and $f$ are two random pixels. Let us assume that the order of pixels in $\mathbb{A}$ and $\mathbb{B}$ are different. This implies according to induction:

$$\exists \quad (a^p, a^q) \in \mathbb{B} \quad \text{s.t.}$$
$$\psi(a^p) > \psi(a^q) \in \mathbb{B} \quad \wedge \quad \psi(a^p) < \psi(a^q) \in \mathbb{A} \tag{10}$$

However, $\psi(a^p) > \psi(a^q) \in \mathbb{B}$ and $\psi(a^p) < \psi(a^q) \in \mathbb{A}$ cannot be true at the same time, we can by mathematical induction deduce that $\nexists \quad (a^p, a^q) \in \mathbb{B}$ that satisfy both conditions given in Equation (10). As such the order of pixels as per their importance are same in both $\mathbb{A}$ and $\mathbb{B}$. We leverage this property that the order of importance of the pixels do not change even in randomly selected (without repetition) subsets for our analysis. If the selected pixels have the same importance ranks, their relative orders are not considered to affect the rank correlation.

## S3 DROP PLOTS FOR ALL DATASETS

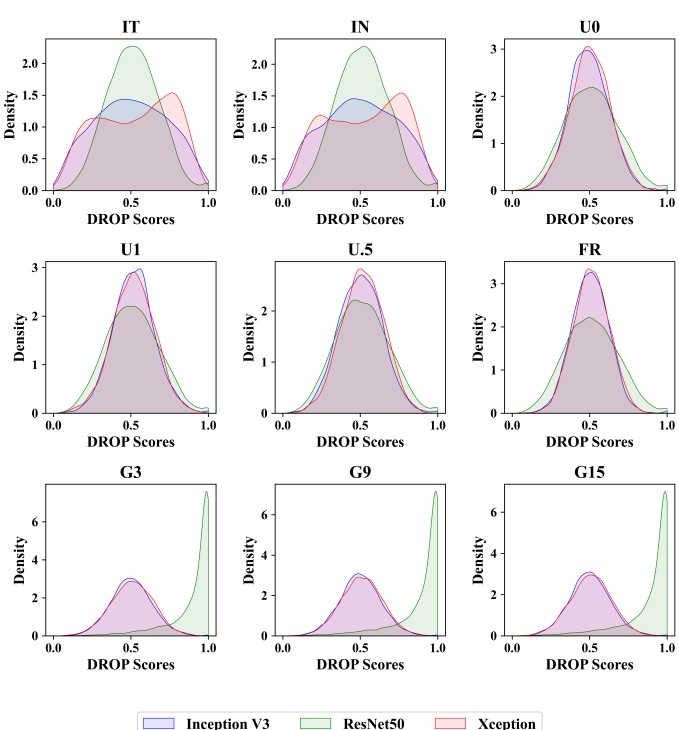

Figure S2: Distribution of $DROP$ scores across all models, perturbation types using pixel-wise perturbation scheme for Imagenette Dataset

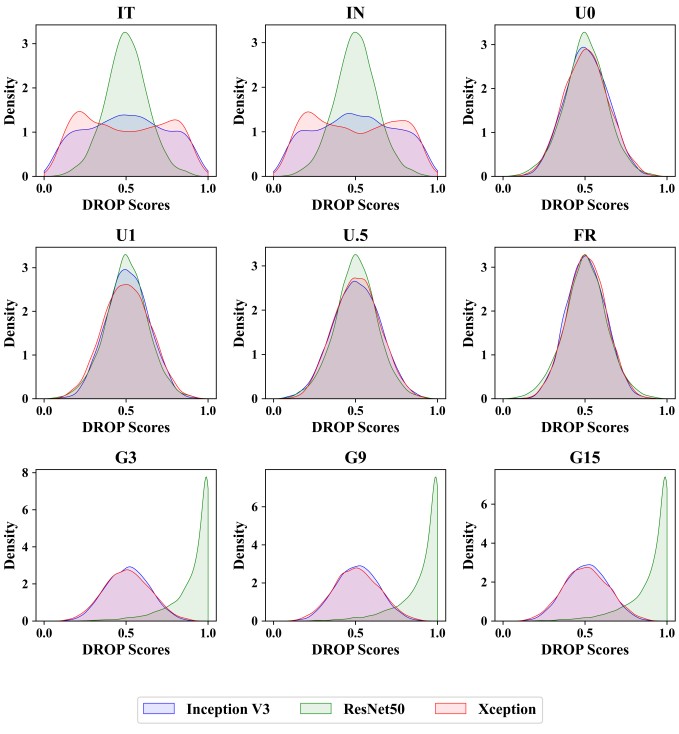

Figure S3: Distribution of $DROP$ scores across all models, perturbation types using pixel-wise perturbation scheme for Oxford-IIIT Pets Dataset

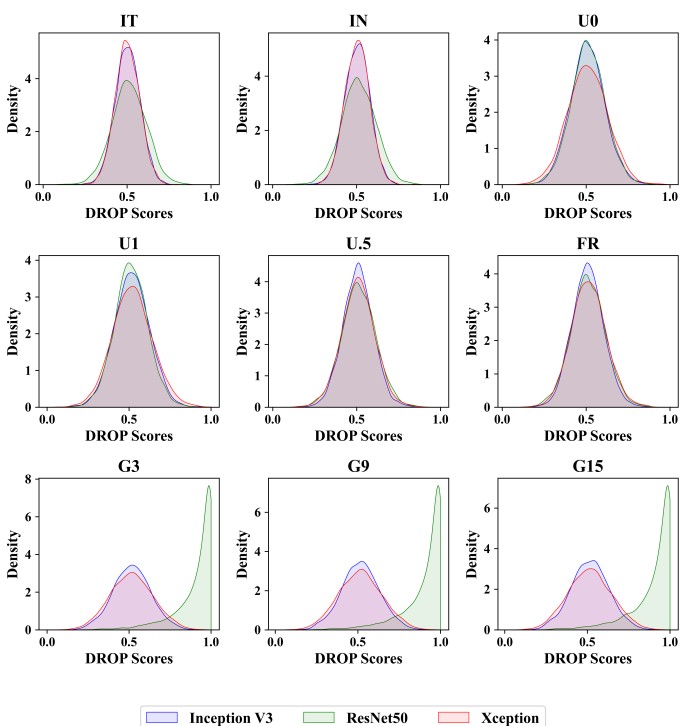

Figure S4: Distribution of $DROP$ scores across all models, perturbation types using pixel-wise perturbation scheme for PASCAL VOC 2007 Dataset

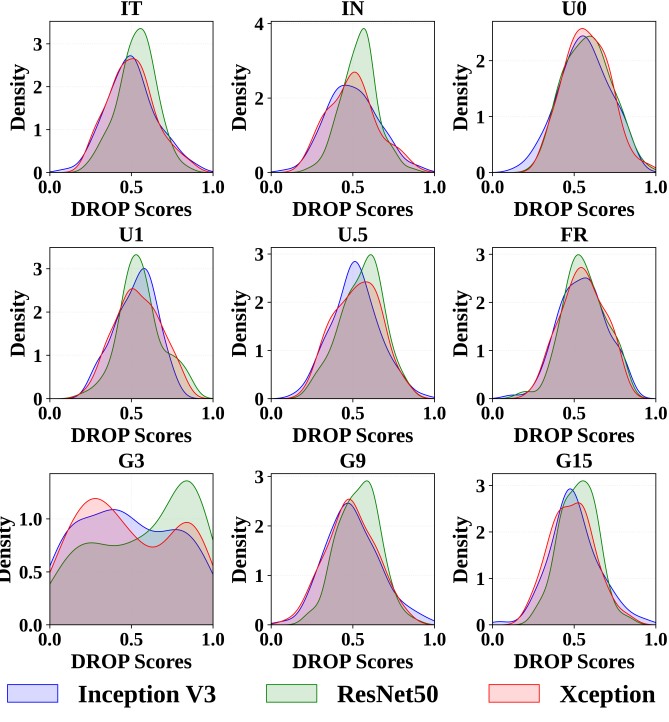

Figure S5: Distribution of $DROP$ scores across all models, perturbation types using segment-wise perturbation scheme for Imagenette Dataset

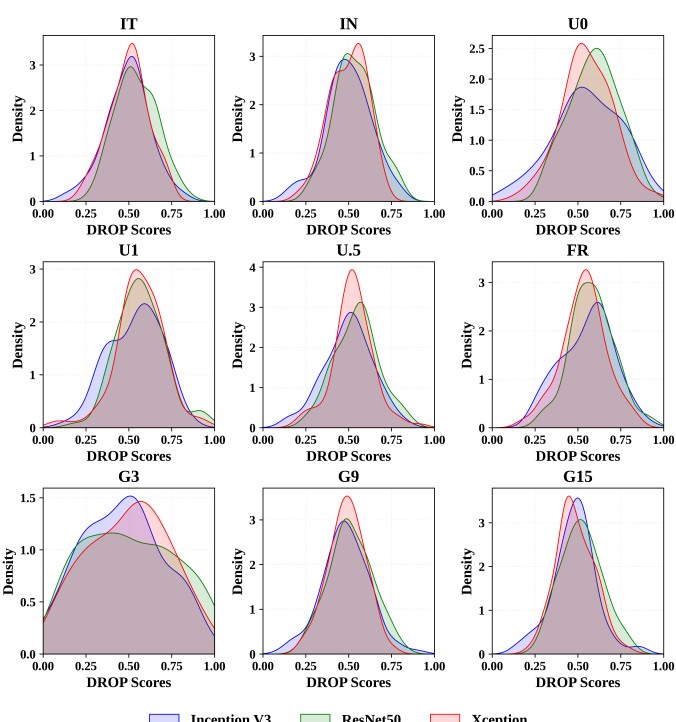

Figure S6: Distribution of $DROP$ scores for all perturbations for Oxford-IIIT Pets Dataset for segment-wise perturbation scheme.

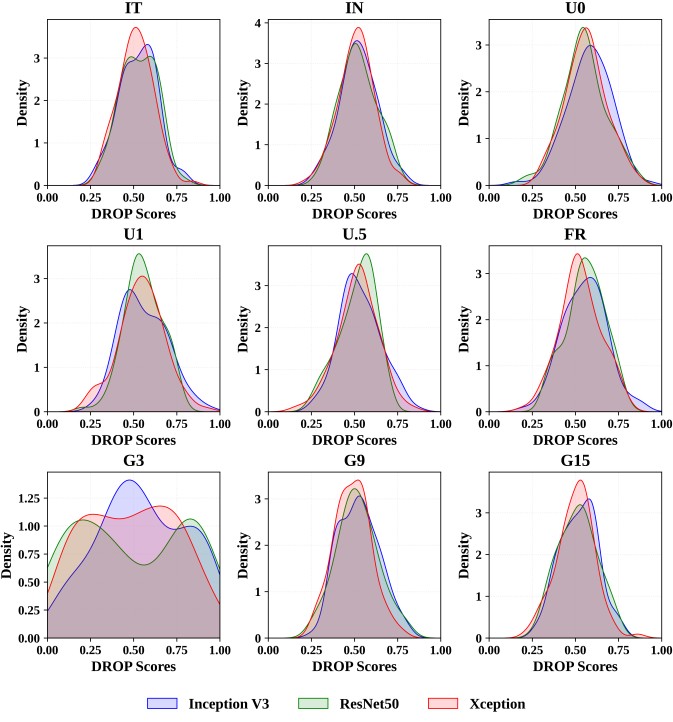

Figure S7: Distribution of $DROP$ scores for all perturbations for PASCAL VOC Dataset for segment-wise perturbation scheme.

## S4  PAIRWISE PSIM PLOTS FOR ALL DATASETS

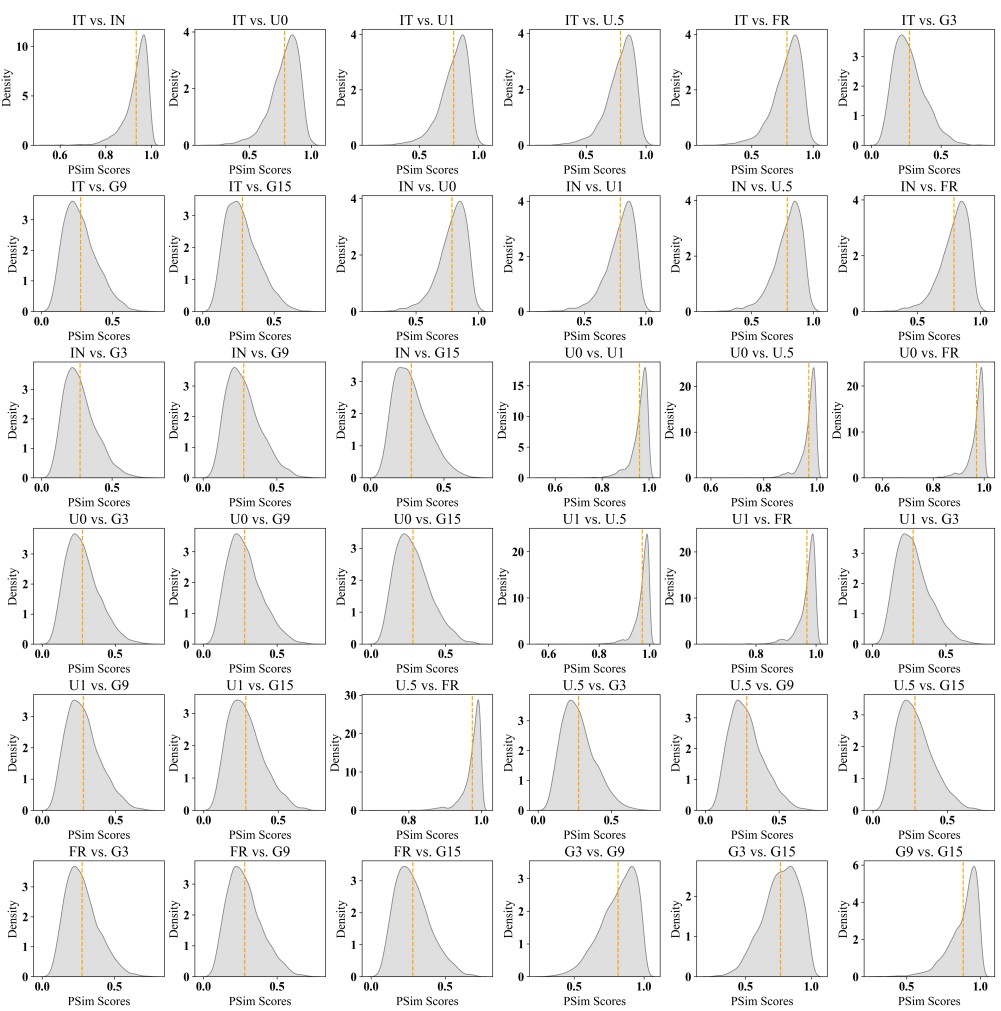

Figure S8: Distribution of pairwise $PSim$ scores for all perturbations for Resnet50 model on Imagenette Dataset for pixel-wise perturbation scheme.

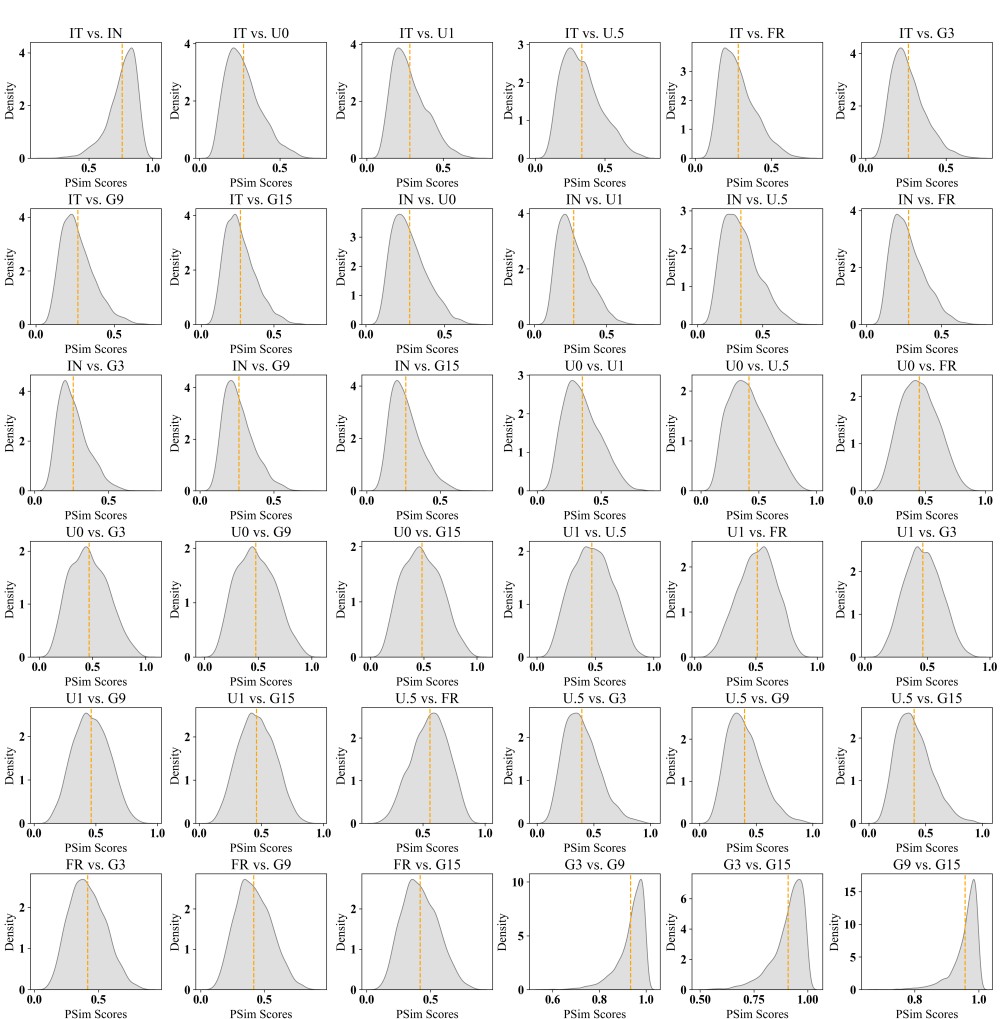

Figure S9: Distribution of pairwise $PSim$ scores for all perturbations for Xception model on Imagenette Dataset for pixel-wise perturbation scheme.

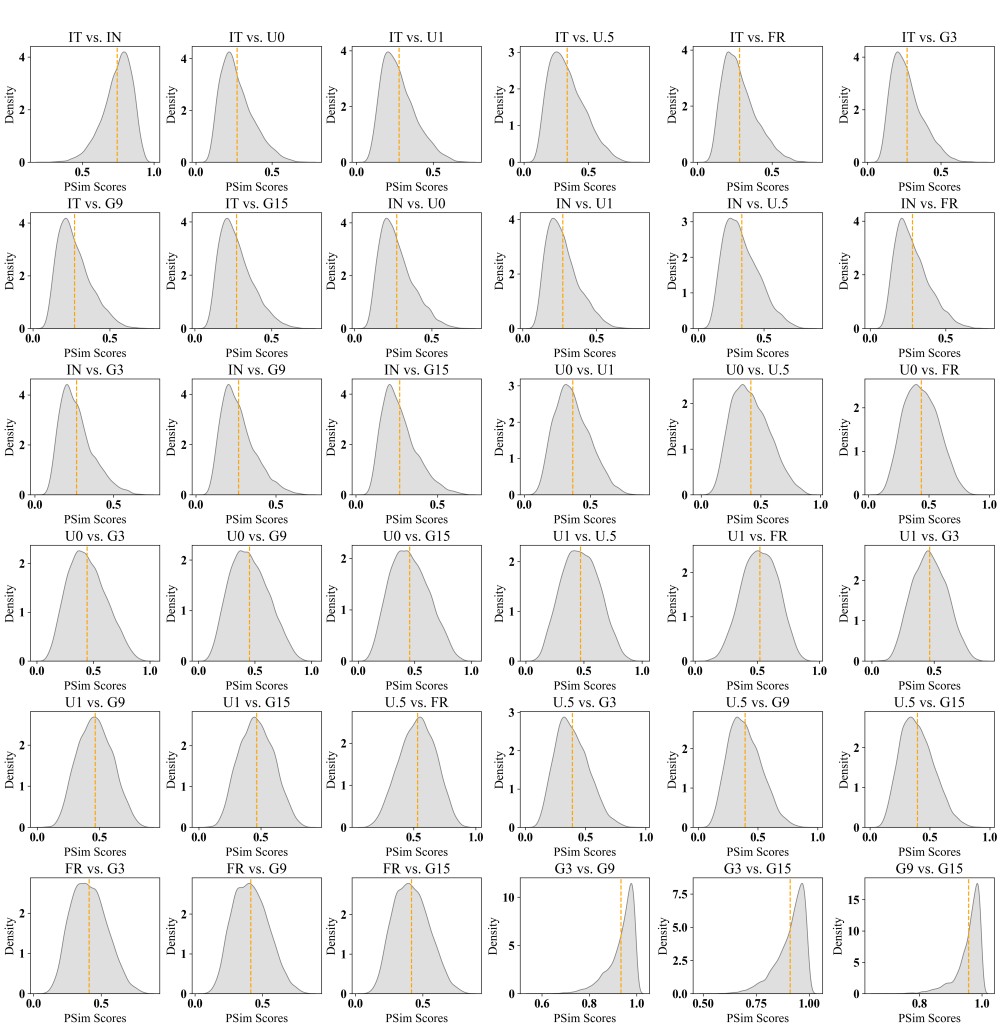

Figure S10: Distribution of pairwise $PSim$ scores for all perturbations for Inception V3 model on Oxford-IIIT Pets Dataset for pixel-wise perturbation scheme.

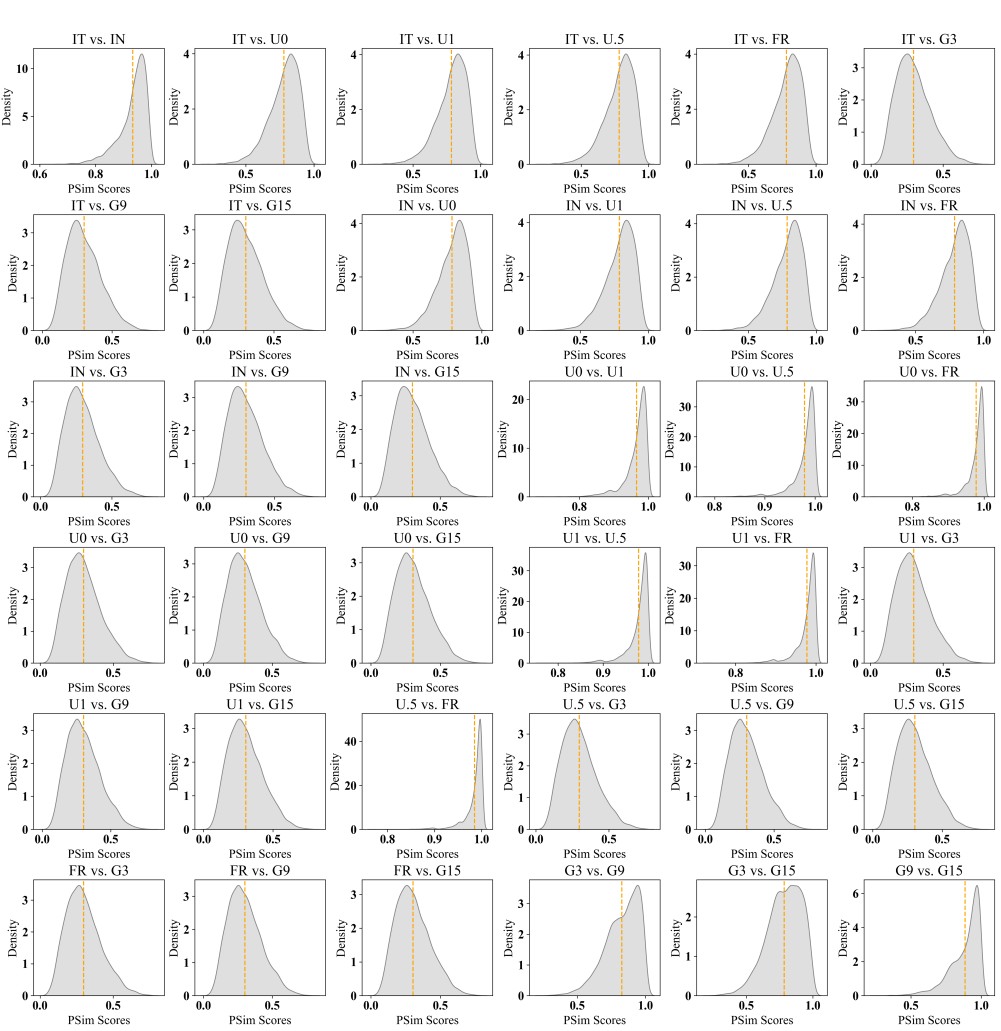

Figure S11: Distribution of pairwise $PSim$ scores for all perturbations for Resnet50 model on Oxford-IIIT Pets Dataset for pixel-wise perturbation scheme.

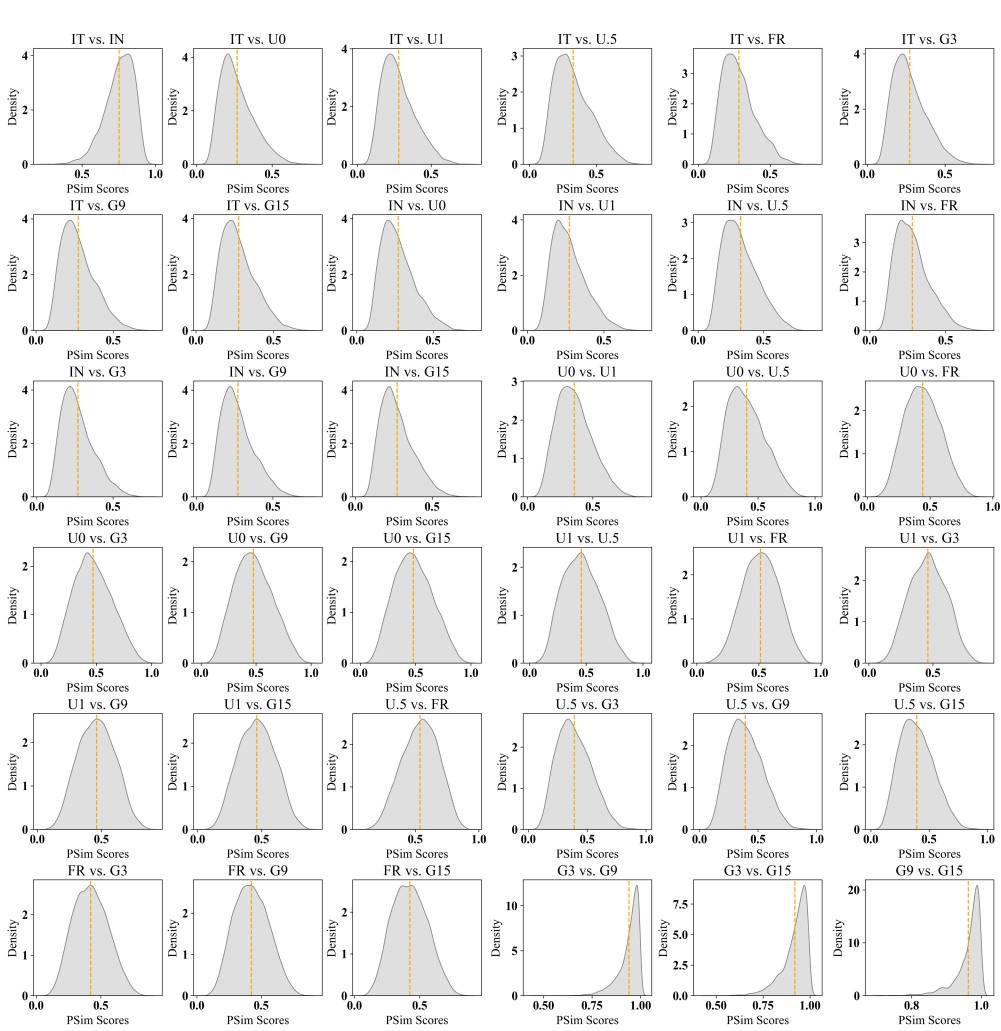

Figure S12: Distribution of pairwise $PSim$ scores for all perturbations for Xception model on Oxford-IIIT Pets Dataset for pixel-wise perturbation scheme.

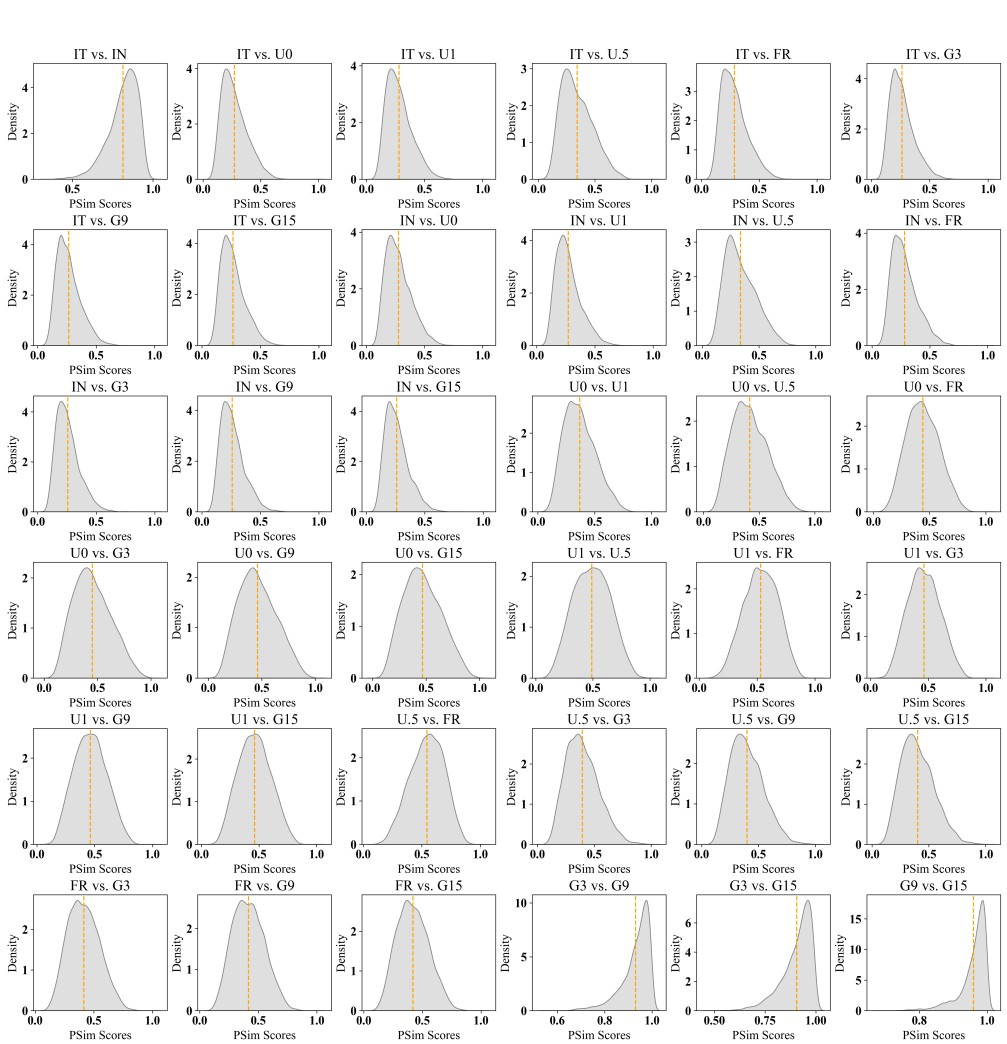

Figure S13: Distribution of pairwise $PSim$ scores for all perturbations for Inception V3 model on PASCAL VOC Dataset for pixel-wise perturbation scheme.

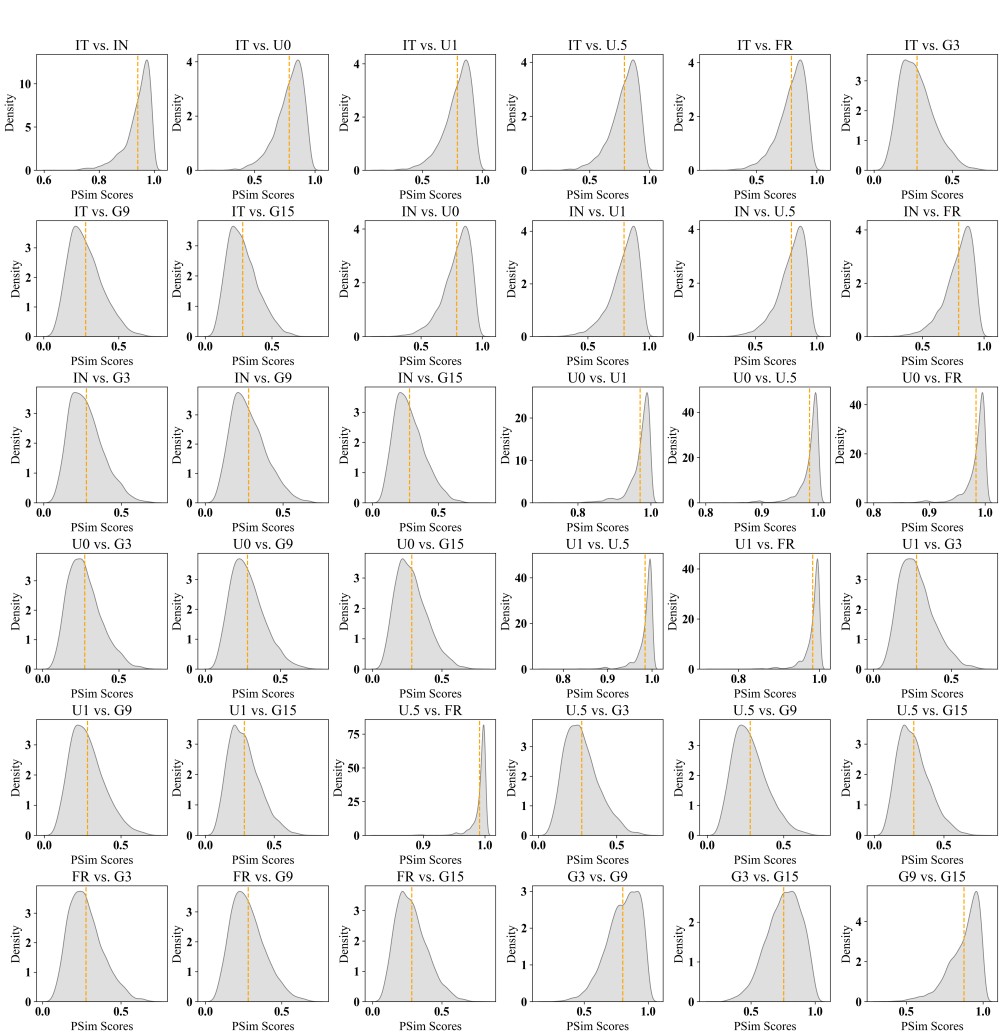

Figure S14: Distribution of pairwise $PSim$ scores for all perturbations for Resnet50 model on PASCAL VOC Dataset for pixel-wise perturbation scheme.

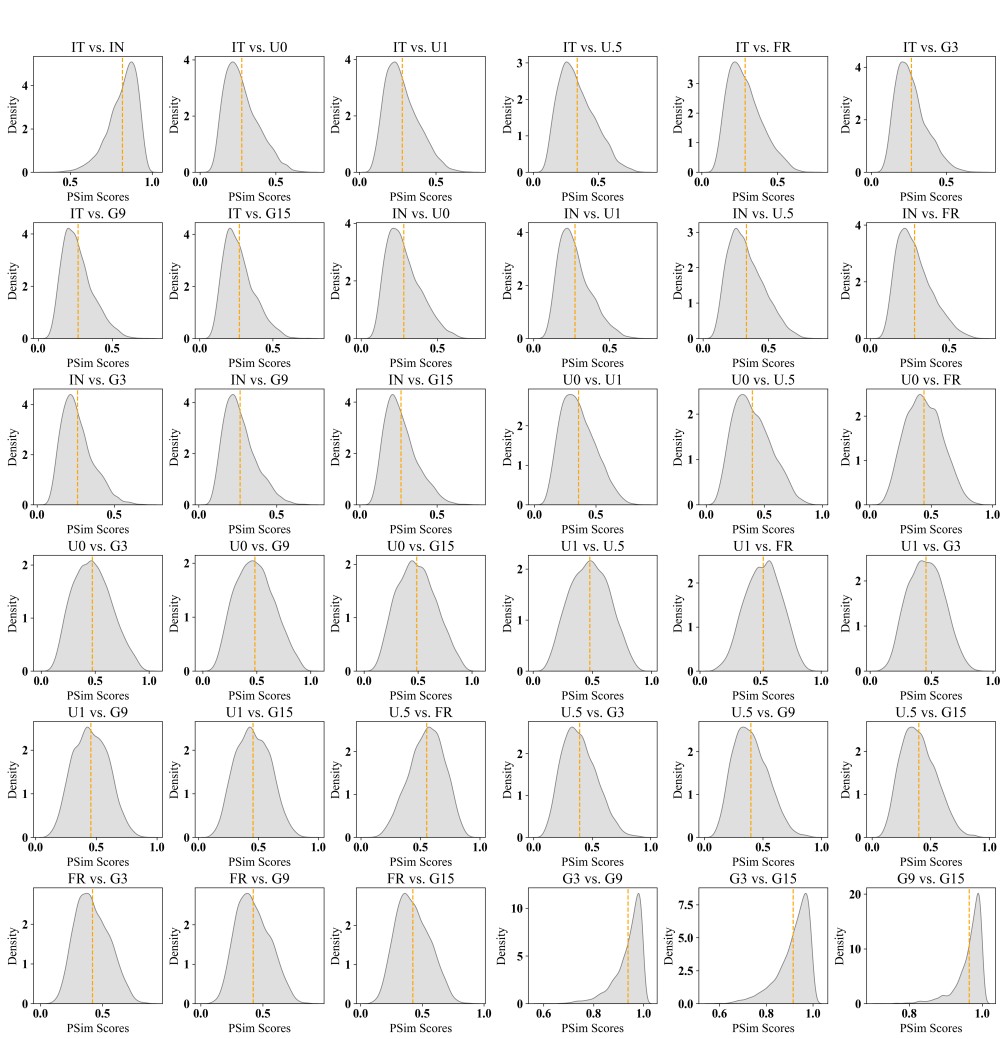

Figure S15: Distribution of pairwise $PSim$ scores for all perturbations for Xception model on PASCAL VOC Dataset for pixel-wise perturbation scheme.

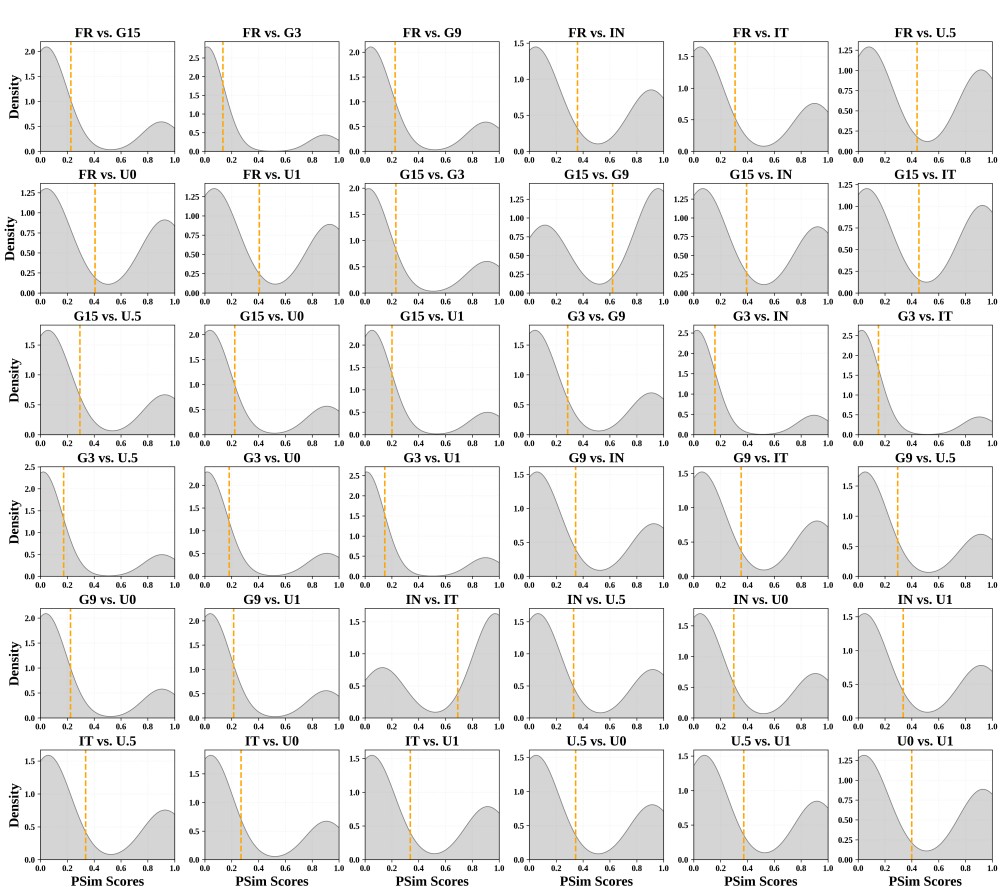

Figure S16: Distribution of pairwise $PSim$ scores for all perturbations for Inception V3 model on Oxford-IIITH Dataset for segment-wise perturbation scheme.

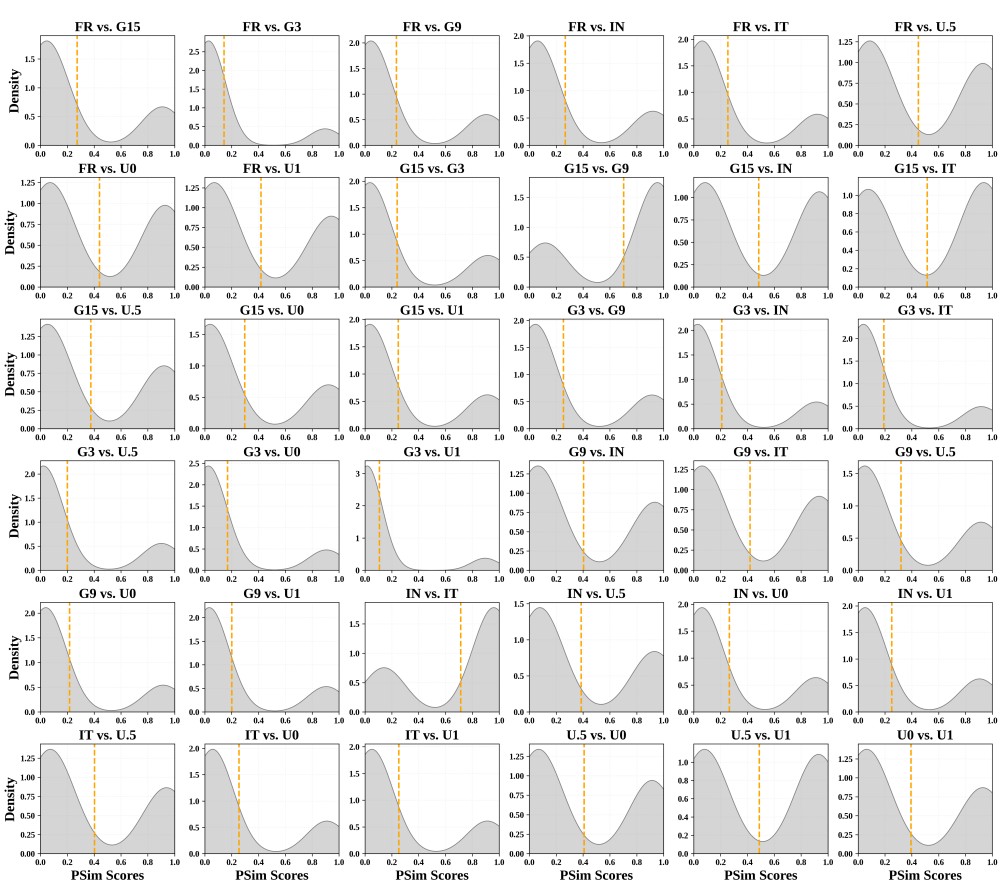

Figure S17: Distribution of pairwise $PSim$ scores for all perturbations for Resnet50 model on Oxford-IIITH Dataset for segment-wise perturbation scheme.

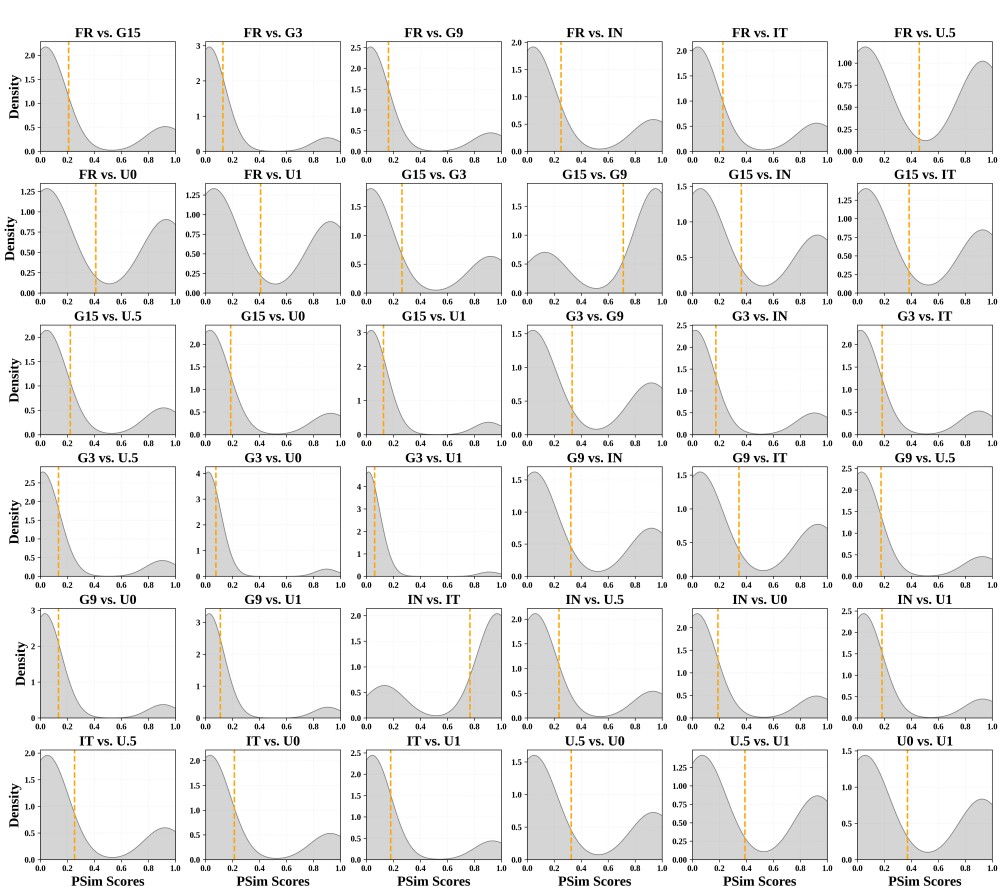

Figure S18: Distribution of pairwise $PSim$ scores for all perturbations for Xception model on Oxford-IIITH Dataset for segment-wise perturbation scheme.

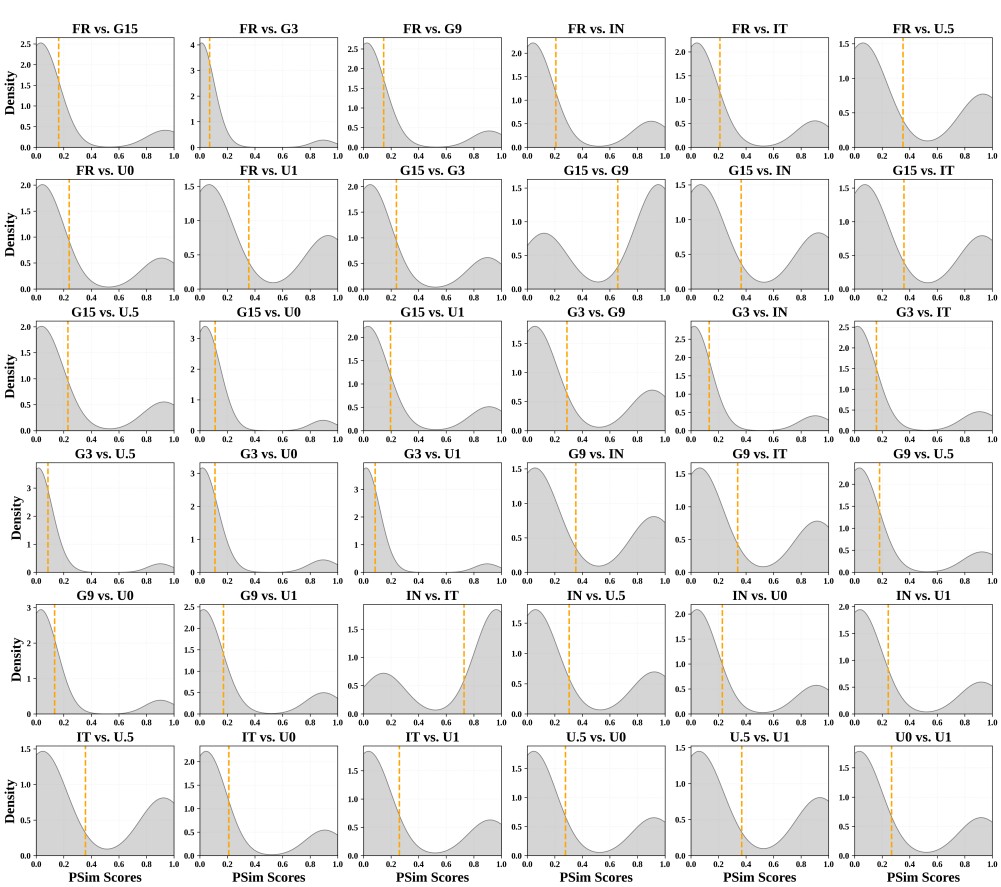

Figure S19: Distribution of pairwise $PSim$ scores for all perturbations for Inception V3 model on Oxford-IIITH Dataset for segment-wise perturbation scheme.

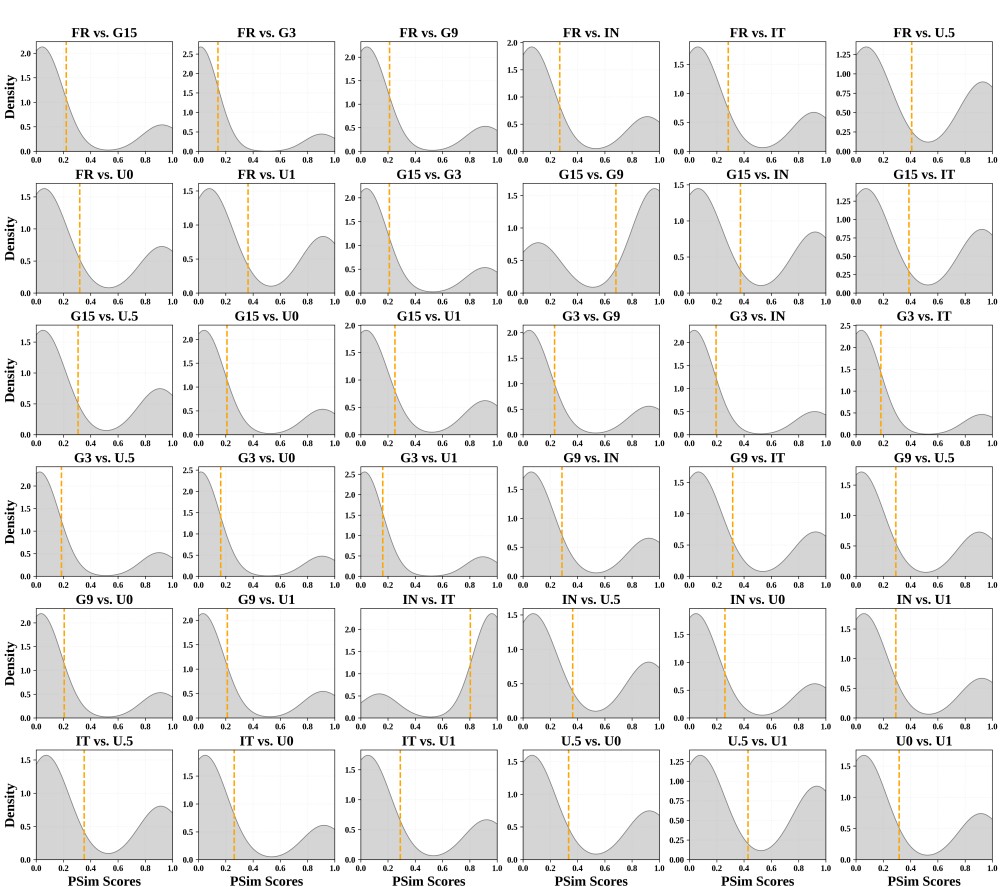

Figure S20: Distribution of pairwise $PSim$ scores for all perturbations for Resnet50 model on Oxford-IIITH Dataset for segment-wise perturbation scheme.

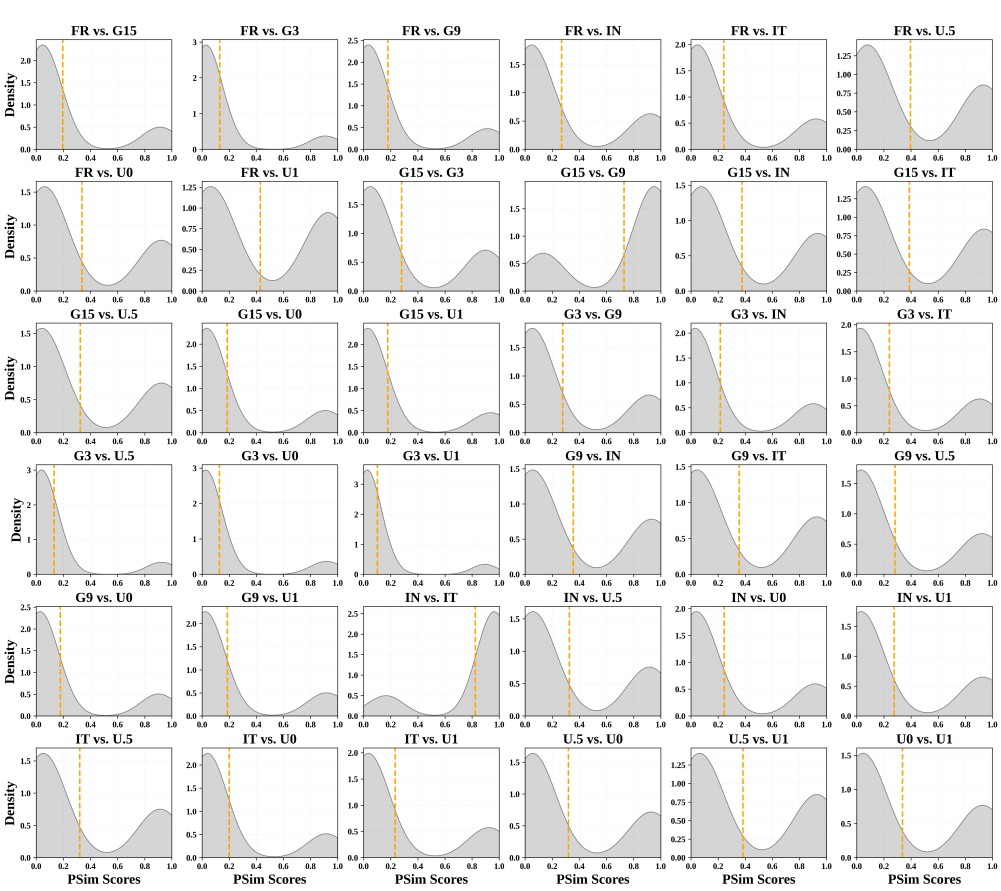

Figure S21: Distribution of pairwise $PSim$ scores for all perturbations for Xception model on Oxford-IIITH Dataset for segment-wise perturbation scheme.

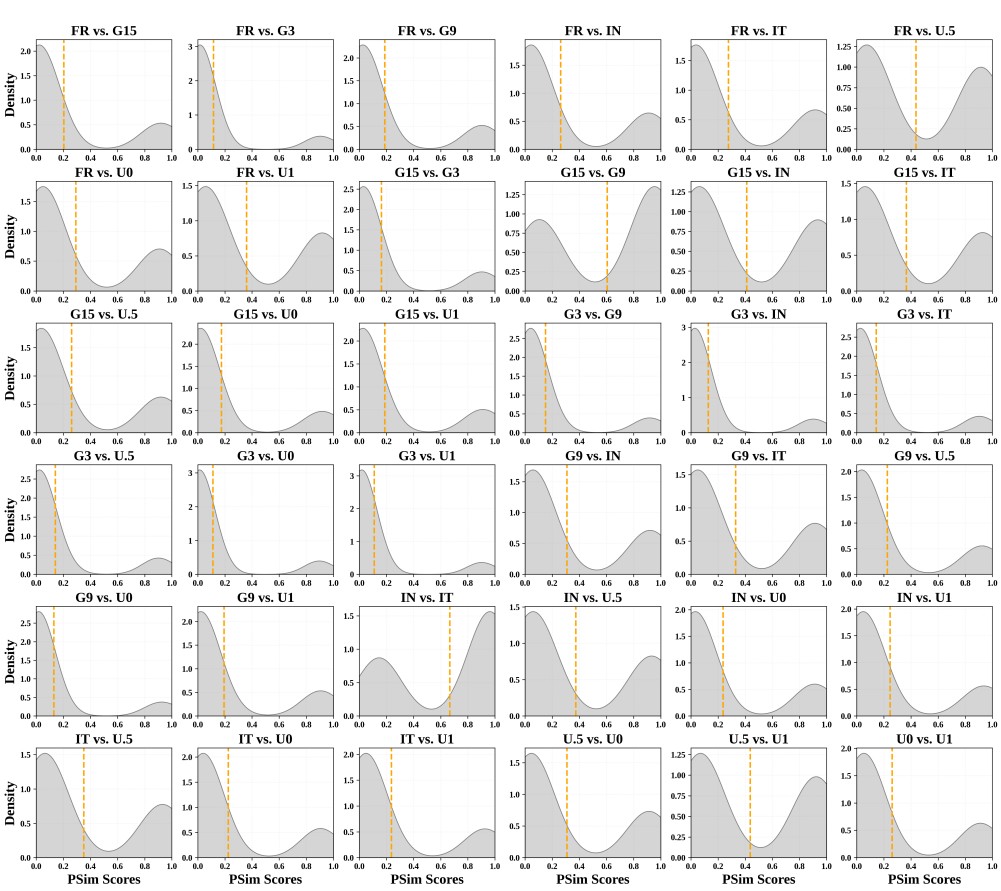

Figure S22: Distribution of pairwise $PSim$ scores for all perturbations for Inception V3 model on PASCAL VOC Dataset for segment-wise perturbation scheme.

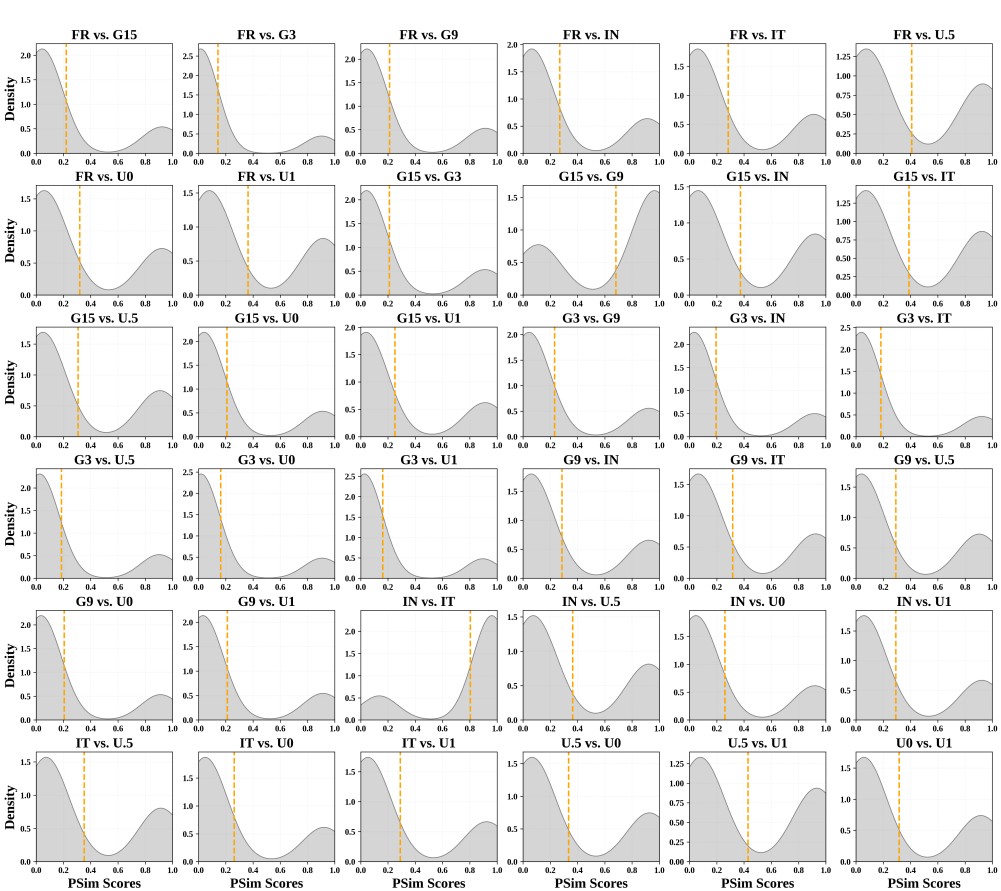

Figure S23: Distribution of pairwise $PSim$ scores for all perturbations for Resnet50 model on PASCAL VOC Dataset for segment-wise perturbation scheme.

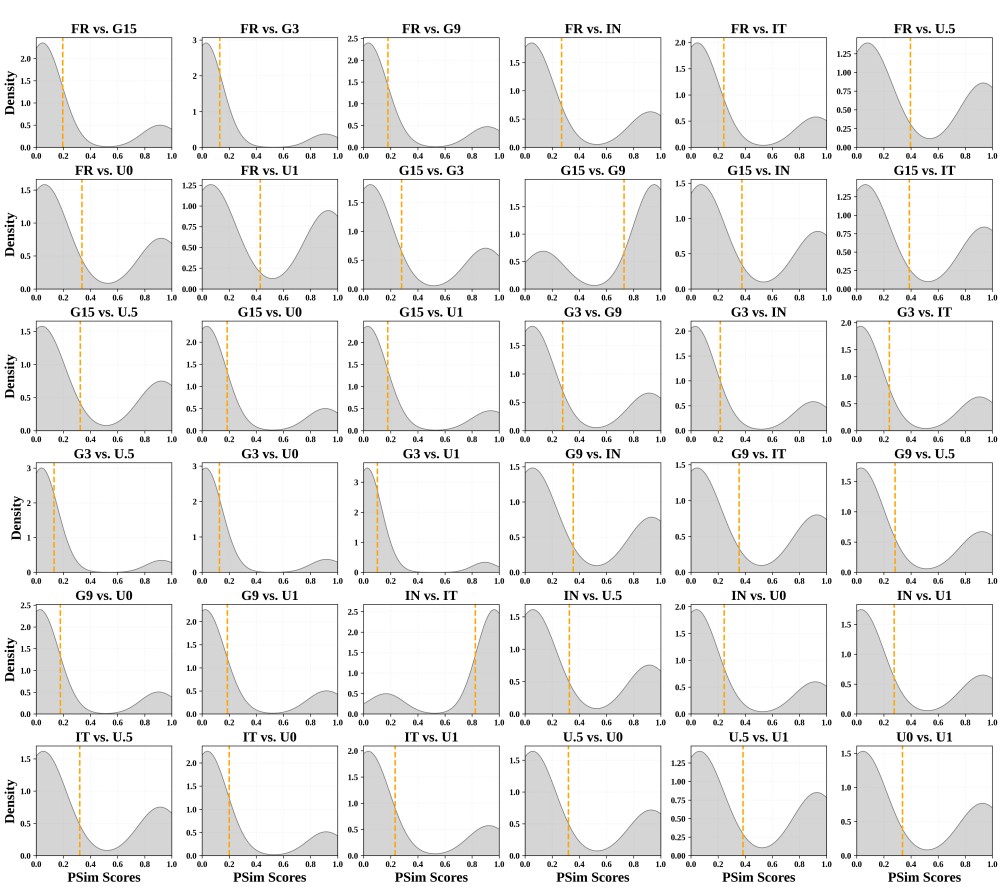

Figure S24: Distribution of pairwise $PSim$ scores for all perturbations for Xception model on PASCAL VOC Dataset for segment-wise perturbation scheme.

# S5 PROBABILITY ESTIMATION OF DROP FOR HIGHER CONFORMITY

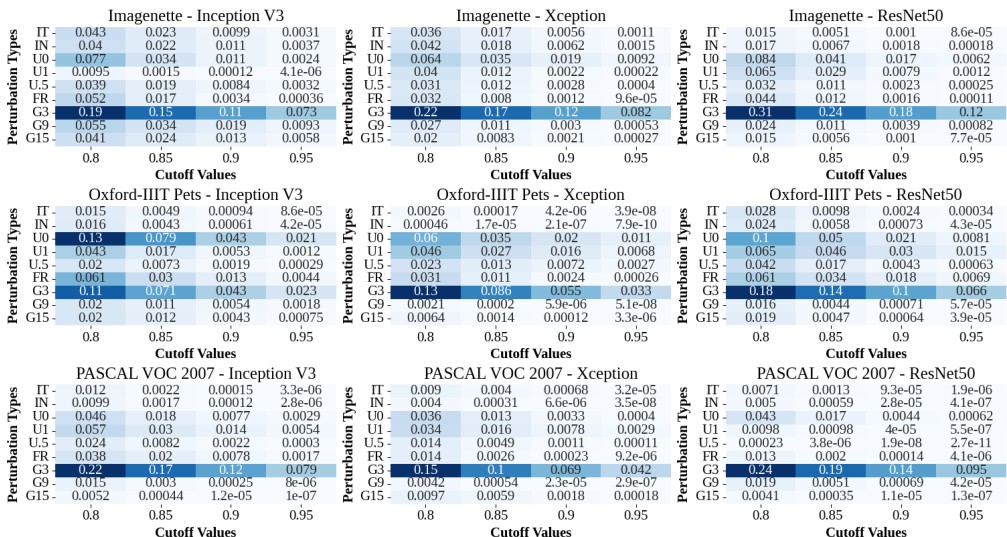

Figure S25: Estimated probabilities of $DROP$ scores for dataset, model, perturbation types using pixel-wise perturbation scheme for different cutoffs.

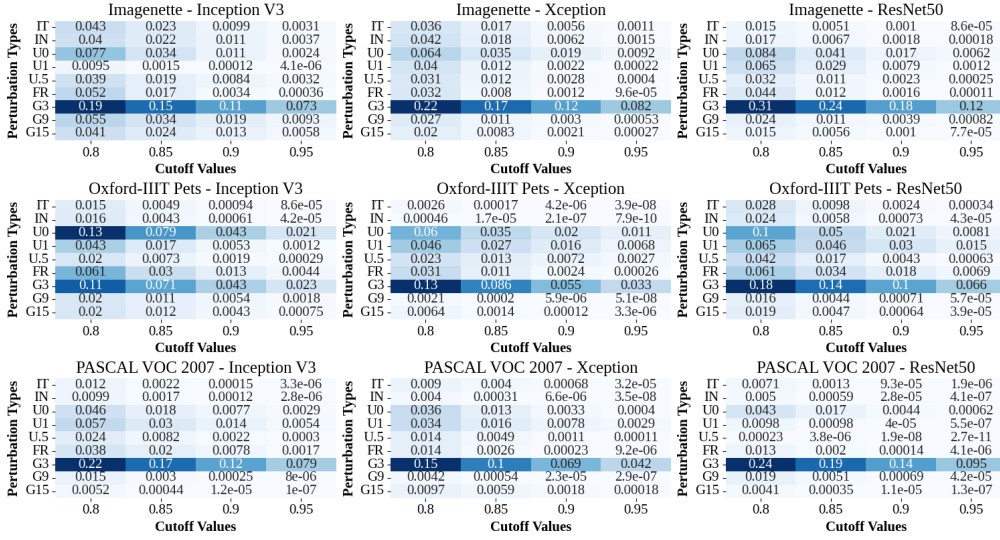

Figure S26: Estimated probabilities of $DROP$ scores for dataset, model, perturbation types using segment-wise perturbation scheme for different cutoffs.

# S6 PROBABILITY ESTIMATION OF PSIM FOR HIGHER CONFORMITY

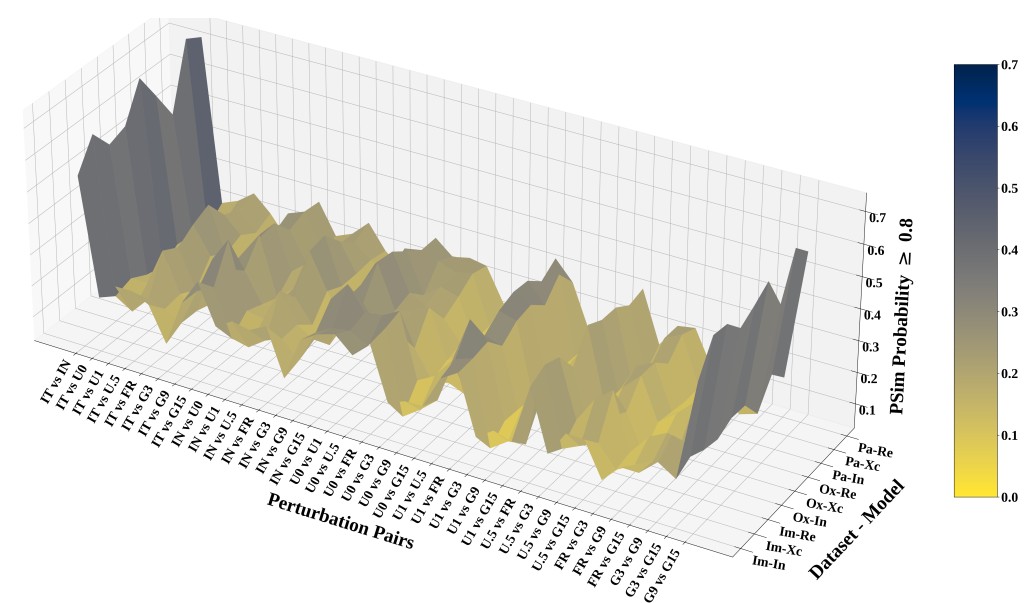

Figure S27: Probabilities of $PSim$ score to be above 0.80 for different perturbation pairs across all dataset:model combinations.

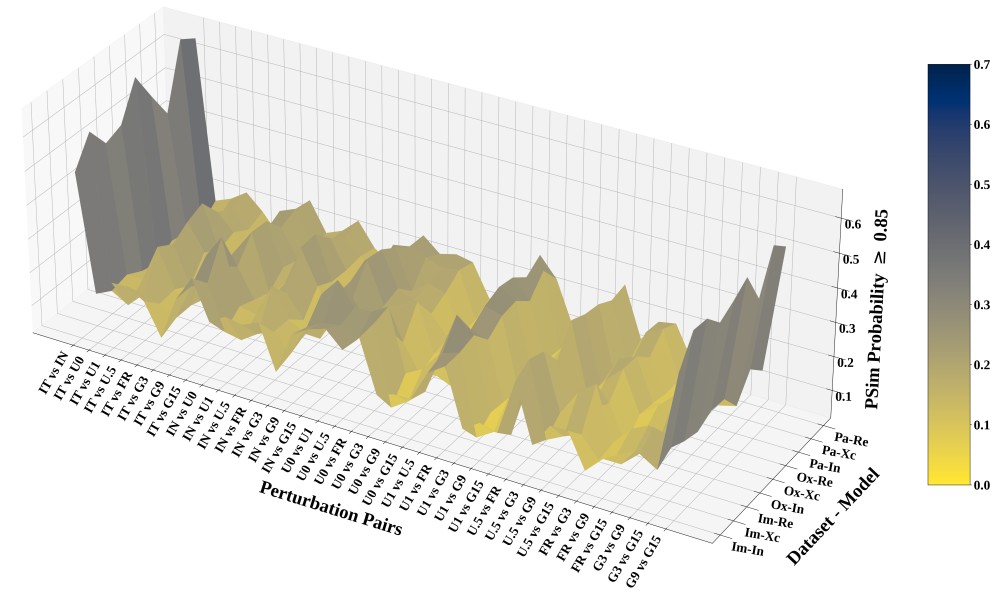

Figure S28: Probabilities of $PSim$ score to be above 0.85 for different perturbation pairs across all dataset:model combinations.

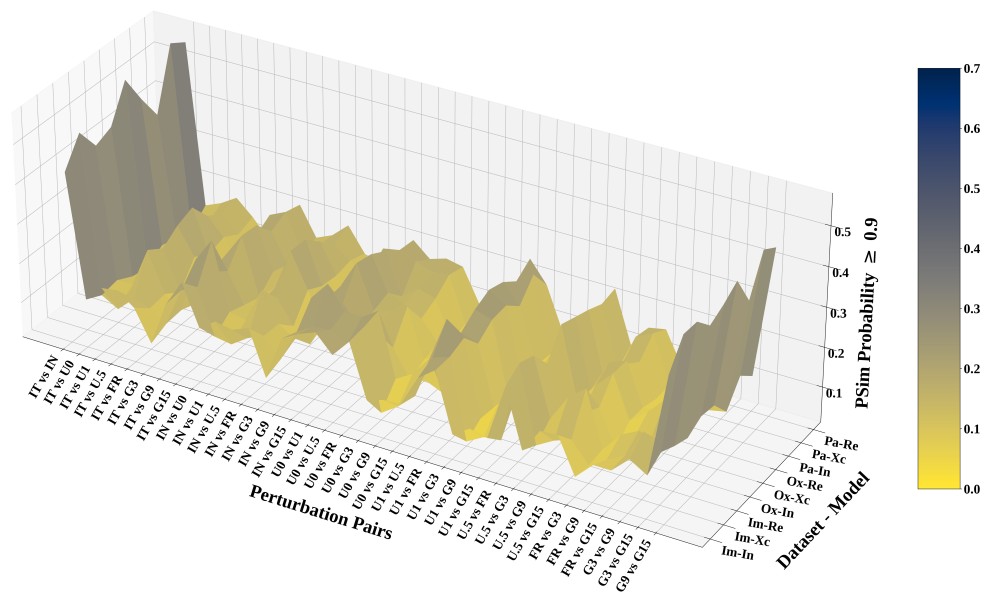

Figure S29: Probabilities of $PSim$ score to be above 0.90 for different perturbation pairs across all dataset:model combinations.

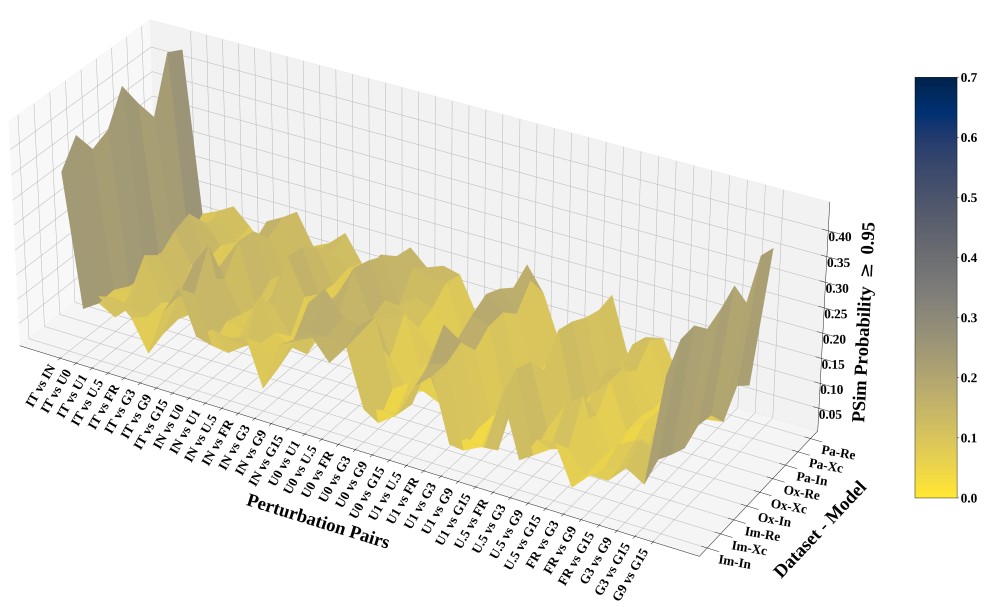

Figure S30: Probabilities of $PSim$ score to be above 0.95 for different perturbation pairs across all dataset:model combinations.

