# OpenReview forum: "Explaining the Inconsistency of Perturbation-Based Fidelity Metrics"
_ICLR.cc/2026/Conference — Submitted to ICLR 2026_

### Official Review · Reviewer_XEKX · 2025-10-21

**Soundness:** 1
**Presentation:** 1
**Contribution:** 1
**Rating:** 2
**Confidence:** 4

**Summary:**

This paper investigates the underlying reasons behind the inconsistency of perturbation-based fidelity metrics used for evaluating saliency maps in deep learning models.  The authors formalize two core assumptions underlying existing fidelity metrics and introduce two new conformity measures, DROP(Drop in Prediction Probability) and PSim(Pixel Rank Similarity), to test those assumptions directly. Experiments are conducted using three pretrained and two adversarially trained models (InceptionV3, Xception, ResNet50) on three datasets (Imagenette, Oxford-IIIT Pets, PASCAL VOC 2007) across nine perturbation types and two perturbation schemes.

**Strengths:**

1. The paper clearly identifies a gap: prior work observed inconsistency or predicted it, but this work seeks to explain it. This shift from "what" to "why" is a valuable contribution.

2. This paper introduces two novel conformity measures, DROP and PSim.  DROP (Eq. 6) quantifies how often perturbations lead to output probability decreases, an interpretable operationalization of P1. PSim (Eq. 7) measures rank invariance across perturbations, directly assessing P2. Both measures provide lightweight diagnostics compared to prior supervised approaches such as FRIES.

3. This paper has an extensive experimental evaluation. It evaluates three pretrained and two adversarially trained models across three datasets and tests nine perturbation types, including inpainting and Gaussian Blur with multiple kernels.  Large-scale computation (≈75 million model predictions) demonstrates strong experimental effort and implementation reproducibility.

**Weaknesses:**

1. Insufficient Justification for Assumptions:
- The paper states that fidelity metrics *assume* "[P1] There is a drop in the output probability when a pixel is perturbed" (Sec 2, line 234) and formalizes this as $p_0 \ge p_i^{\phi}$ (Eq 2). This assumption seems overly strong and is not well-defended. It is plausible that perturbing (e.g., masking) a pixel that provides *negative* or *distracting* evidence for the correct class could *increase* the output probability.
- There is no evidence to support these two assumptions. This paper also does not claim the relationship between these two assumptions and metric faithfulness. It never establishes a theoretical or empirical link between satisfying these assumptions and a fidelity metric’s ability to measure true faithfulness.

2. The approach setting:
- In Algorithm 1, this paper calculates the metric DROP and PSim via perturbing pixel by pixel. In my opinion, it tests the robustness of the pretrained model rather than measuring the faithfulness. For one picture, only one or two pixels removed or added to the noise, the predictions are supposed to have a minor change.
- In Eq. 5, this paper claims that the ideally Rank Biased Overlap should be one. In Eq. 7, this paper introduces different perturbations. However, previous papers demonstrate the out-of-distribution problem should be considered. For example, when a segmentation was removed in this picture, it is hard to tell if the prediction is affected by OOD problem or the faithfulness.
- This paper uses CNN-based architectures. It is better to introduce other architecures, such as transformer, LSTM, etc.

3. The presentation:
- The citations are not consistent. In line 35,36, the citation is the same as the main body, making it hard to read. In line 61, 63, 70, two other style citations are introduced.
- Inconsistent terms. For example, in line 178, 189, there are two style **DROP**. In line 195,203, there are two style **PSim**.
- The algorithm's readability can be improved.
- The notation in Equation (1) ($\mathfrak{R}=\{a_{1},a_{2},a_{3},a_{4},...a_{i}\}$) and its description ("$a_{1}\rightarrow a_{i}$ are pixels sorted... a greater i denotes greater importance") is confusing. It seems to mix indices ($i$) with pixel identifiers ($a_i$). It is unclear if $i$ is the total number of pixels or an index.

4. Limited reference:
- The references used in this paper were mainly published before 2021. In the related works, only two papers were mentioned. Some important papers are missing, such as ROAR[1], ROAD[2], F-Fidelity[3].

Reference:
- [1].  A benchmark for interpretability methods in deep neural networks.NIPS 2019.
- [2].  A Consistent and Efficient Evaluation Strategy for Feature Attributions. ICML 2022
- [3]. F-Fidelity: A Robust Framework for Faithfulness Evaluation of Explainable AI. ICLR 2025

**Questions:**

- Why is it assumed that perturbing *any* pixel should *drop* the output probability? Couldn't perturbing (e.g., masking) a pixel that provides *negative* evidence for the correct class *increase* the probability?
-  How would these results differ for transformer-based or self-attention models?
- Are there cases where low conformity does *not* imply low fidelity reliability?
- How generalizable are the conclusions to non-vision modalities or structured data?
- Could conformity measures be incorporated directly into the training objective to improve interpretability?

---

> ### Author Response · Authors · 2025-12-04
> **Authors' rebuttal to Reviewer XEKX (part 1)**
>
> We thank the reviewer for the comments. Our comments regarding the weaknesses, Questions posted by reviewer are as below:
>
> **W1:** "Insufficient Justification for Assumptions: The paper states that fidelity metrics assume "[P1] There is a drop in the output probability when a pixel is perturbed" .... "
> **Rebuttal:** We would like to draw the attention of reviewer to the role of [P1] as already explained in both the Introduction and Section 2 of the submitted paper: [P1] is **not a behavioral claim about deep models, but the validity condition implicitly required by fidelity metrics** such as AOPC, Faithfulness, Insertion/Deletion AUC etc. These metrics only function meaningfully under the assumption that removing relevant evidence leads to a drop in the model’s output probability. This assumption is also made explicitly in FRIES (Bora et al., Pattern Recognition 2026). The reviewer’s counterexample i.e., removing a negatively contributing pixel increasing probability is therefore not a challenge to our assumption, contrarily is a prime example of how fidelity metrics’ assumption is not true, and DROP is designed to detect.
>
> **W2:** "Insufficient Justification for Assumptions: There is no evidence to support these two assumptions. This paper also does not claim the relationship between ...."
> **Rebuttal:** **The link between assumptions [P1]/[P2] and fidelity-metric validity is already established in Section 2 of the submitted paper**. These assumptions are not proposed by us, but they are the implicit requirements of fidelity metrics such as AOPC, Insertion/deletion AUC and Faithfulness etc. As mentioned in Section 1.1, **this relationship is explicitly used in prior work FRIES (Bora et al., Pattern Recognition 2026), and implicitly in Samek et al. (TNNLS 2017)**, rely on these assumptions when interpreting fidelity scores. Our contribution is to formalize these assumptions and empirically test whether they actually hold. Thus, the reviewer’s claim that we do not establish the connection is based on a misunderstanding: the fidelity metrics themselves define faithfulness using [P1] and [P2], and our results show that these required behaviors are frequently violated in practice.
>
> **W3:** "The approach setting: In Algorithm 1, this paper calculates the metric DROP and PSim via perturbing pixel by pixel. ...."
> **Rebuttal:** **DROP and PSim are not intended to measure faithfulness of explanations but they measure whether the validity assumptions of fidelity metrics hold for a given model–perturbation pair or not.** Fidelity metrics such as AOPC, Insertion/Deletion AUC require that even small evidence removals produce monotonic and proportional probability changes (**as explained in details in Section 1 and Section 2 of the paper**), because they aggregate these local effects into their global scores. Prior work (e.g., Tomsett et al., FRIES mentioned in Section 1.1) also evaluates perturbations at fine granularity for this reason. Violations at the pixel level already indicate that the fidelity metric’s core premises do not hold.
>
> **W4:** "The approach setting: In Eq. 5, this paper claims that the ideally Rank Biased Overlap should be one..... "
> **Rebuttal:** We agree that certain perturbations can push images outside the natural data manifold. Since existing fidelity metrics operate under this same perturbation protocol and do not correct for such OOD drift, we follow the standard setup to remain consistent with their assumptions. Thus, OOD effects are not confounding the results rather they are the mechanism that fidelity metrics fail to address, and DROP/PSim highlight and quantify this failure in a principled way.
>
> **W5:** "The approach setting: This paper uses CNN-based architectures. ..... "
> **Rebuttal:** We focus on CNN-based architectures because deletion-based fidelity metrics were originally defined and are predominantly used in vision models built on CNNs. This setting also exposes the core problem most sharply: in images, “deletion’’ cannot be performed cleanly and may induce OOD artifacts, which is exactly what our diagnostics are designed to reveal. Since DROP and PSim operate on the metric–perturbation interaction rather than on architectural specifics, the conclusions are not tied to CNNs.
>
> **W6:** "The presentation: Citations and Terms "
> **Rebuttal:** We made the citations and terms consistent in the revised version.
>
> **W7:** "The presentation: The notation in Equation (1) () and its description (" are pixels sorted... a greater i denotes greater importance") is confusing...."
> **Rebuttal:** In Eq. (1): R={a_1,a_2,…,a_i},
> a_i  denotes the pixel occupying rank iin the saliency ordering, and i is simply the rank index, while N is the total number of pixels. Since the subsequent analysis depends only on the ordering and not on pixel identities, we use a single index for clarity.

---

> ### Author Response · Authors · 2025-12-04
> **Authors' rebuttal to Reviewer XEKX (part 2)**
>
> **W8: Limited reference: The references used in this paper were mainly published before 2021. In the related works, only two papers were mentioned. Some important papers are missing, such as ROAR[1], ROAD[2], F-Fidelity[3].**
> **Rebuttal:** The reviewer’s suggested papers: ROAR, ROAD, and F-Fidelity, are indeed influential works on evaluating saliency methods using fidelity metrics, but they are not the closest references to our contribution. These works introduce or benchmark new evaluation procedures, whereas our paper investigates the validity assumptions underlying fidelity metrics themselves. The most directly related prior works, as mentioned in Section 1.1, are Tomsett et al. (AAAI 2020), which first reported unexplained inconsistency across perturbations, and FRIES (Bora et al., Pattern Recognition 2026), which estimates this inconsistency by building supervised model. Our paper builds on these closest precedents by formalizing the assumptions (P1, P2) required by fidelity metrics and diagnosing where they fail. We did not cite ROAR, ROAD, and F-Fidelity as they address a different problem.
>
> **Q1:** "Why is it assumed that perturbing any pixel should drop the output probability? ...."
> **Rebuttal:** We respectfully clarify that the role of [P1] is already explained in both the Introduction and Section 2 of the submitted paper: **[P1] is not a behavioral claim about deep models, but the validity condition implicitly required by fidelity metrics** such as AOPC, Faithfulness, Deletion AUC etc. These metrics only function meaningfully under the assumption that removing relevant evidence leads to a drop in the model’s output probability (i.e. condition P1). This assumption is also made explicitly in FRIES (Bora et al., Pattern Recognition 2026). **The reviewer’s counterexample i.e., removing a negatively contributing pixel increasing probability is therefore not a challenge to our assumption, but precisely an example of how fidelity metrics’ assumption is not true, and DROP is designed to detect and quantify it.**
>
> **Q2:** "How would these results differ for transformer-based or self-attention models?"
> **Rebuttal:** We focus on CNN-based architectures because deletion-based fidelity metrics were originally defined and are predominantly used in vision models built on CNNs. This setting also exposes the core problem most sharply: in images, “deletion’’ cannot be performed cleanly and may induce OOD artifacts, which is exactly what our diagnostics are designed to reveal. Since DROP and PSim operate on the metric–perturbation interaction rather than on architectural specifics, the conclusions are not tied to CNNs.
>
> **Q3:** "•	Are there cases where low conformity does not imply low fidelity reliability?"
> **Rebuttal**: DROP and PSim evaluate whether the validity assumptions that fidelity metrics rely on i.e., monotonicity and proportionality under perturbation are satisfied. If these assumptions fail, the mathematical premises of the fidelity metrics no longer hold (as explained in details in Section 1 and Section 2), and their scores become uninterpretable. Therefore, a setting where DROP/PSim are low but the fidelity metric remains “reliable” cannot occur: low conformity means that the conditions required for the metric to function correctly are violated.
>
> **Q4:** "How generalizable are the conclusions to non-vision modalities or structured data?"
> **Rebuttal:** Our analysis focuses on vision because fidelity metrics behave fundamentally differently in images: individual pixels cannot be “removed,” only modified, and such modifications often introduce off-manifold artifacts. This ambiguity is precisely what makes fidelity metrics fragile in the vision setting. In non-vision modalities such as tabular or structured data, feature deletion is well-defined (e.g., dropping a column), and the perturbation mechanism does not suffer from the same ambiguity. Our conclusions therefore target the domain where the problem is most severe and most widely reported, while the diagnostic formulation itself remains modality-agnostic.
>
> **Q5:** "Could conformity measures be incorporated directly into the training objective to improve interpretability?"
> **Rebuttal:** Thank you for this insightful suggestion. In principle, conformity measures such as DROP and PSim could indeed be incorporated into training as regularizers that encourage models to satisfy the validity assumptions required by fidelity metrics. Our work positions DROP and PSim as diagnostic tools, but it does not prevent them from being used as training objectives. Such a direction would correspond to learning models whose perturbation behavior is explicitly constrained toward P1 and P2, and may lead to architectures with inherently more consistent explanations.We agree this is a promising direction for future research.

---

### Official Review · Reviewer_rfHB · 2025-10-27

**Soundness:** 3
**Presentation:** 2
**Contribution:** 2
**Rating:** 4
**Confidence:** 4

**Summary:**

This paper investigates why perturbation-based fidelity metrics used for evaluating saliency maps, such as AOPC, Average Drop, and Faithfulness, often yield inconsistent results across perturbation types.
Rather than proposing a new metric, the authors aim to explain the causes of these inconsistencies by formalizing the implicit assumptions behind such metrics.
They introduce two measures: DROP (Drop in Prediction Probability) and PSim (Pixel Rank Similarity), which test whether model outputs conform to the expected behavior under perturbation.
Experiments across multiple architectures (InceptionV3, Xception, ResNet50, and adversarial variants) and datasets (Imagenette, Oxford-IIIT Pets, PASCAL VOC 2007) reveal widespread violations of these assumptions.
The authors conclude that fidelity evaluations should always report the perturbation type used and test conformity beforehand.

**Strengths:**

- The paper addresses an underexplored question: why fidelity metrics fail rather than how much they fail. This conceptual shift from measurement to explanation is valuable for the XAI community.

- The proposed DROP and PSim scores provide lightweight, interpretable diagnostics that can be computed without training new models, offering a practical tool for assessing metric reliability.

- The experiments span several CNN architectures, adversarial training, and diverse perturbation types, demonstrating the pervasiveness of inconsistency. The large-scale empirical study adds credibility to the conclusions.

**Weaknesses:**

- The formalization of assumptions (P1, P2) is clear but descriptive. The paper does not deeply analyze why certain perturbations, such as Gaussian blur, produce higher conformity, or what model characteristics drive inconsistency.

- While DROP and PSim are useful, they remain diagnostic tools rather than conceptual breakthroughs. Related studies, such as Schulz et al. (Restricting the Flow: Information Bottlenecks for Attribution, ICLR) and Šimić et al. (Perturbation Effect, CIKM 2022), have already examined the robustness and validation of attribution metrics under perturbations, suggesting that this work extends rather than advances the discussion.

- The study involves tens of millions of forward passes but largely confirms intuitive expectations that fidelity metrics depend on perturbation choice, with limited new explanatory mechanisms offered.

- The work is restricted to image-based saliency metrics, whereas recent studies have explored perturbation effects and attribution consistency more extensively in time-series data. This limits generalizability and leaves open whether the same inconsistencies hold in sequential or temporal domains.

**Questions:**

- Why is Gaussian blur the most consistent perturbation type? Does this reflect properties of convolutional inductive biases or feature smoothness? A more theoretical explanation would be valuable.

- How do DROP/PSim scores correlate with human-annotated explanation quality, if at all? Even a small user study or qualitative check could contextualize the practical meaning of “consistency.”

- Would the same inconsistency patterns appear in non-visual modalities (e.g., text models or tabular classifiers)? Extending beyond image classification could greatly increase impact.

- Given the massive computational load, can DROP/PSim be approximated efficiently (e.g., via sampling strategies or low-rank perturbation models)?

---

> ### Author Response · Authors · 2025-12-04
> **Authors' rebuttal to Reviewer  rfHB (part 1)**
>
> We thank the reviewer for the comments. Our comments regarding the weaknesses, Questions posted by reviewer are as below:
>
> **W1:** "The formalization of assumptions (P1, P2) is clear but descriptive"
> **Rebuttal:** Gaussian blur shows higher conformity because it perturbs the image in a smooth, low-frequency manner that avoids the sharp artifacts introduced by masking, noise, or inpainting. Such smooth perturbations tend to keep images closer to the natural-image manifold thereby preserving neighborhood, which aligns with well-known convolutional inductive biases and explains the more stable probability responses. While a full theoretical analysis of these perturbation–model interactions is beyond the scope of the present work, our empirical results consistently indicate that smoother operators behave more reliably than harsher perturbations.
>
> **W2:** "While DROP and PSim are useful, they remain diagnostic tools rather than conceptual breakthroughs."
> **Rebuttal:** Šimić et al. introduce Perturbation Effect Size as a corrective measure for misleading fidelity evaluation in time-series models, showing that selectively perturbing only high-relevance features can distort validation. Schulz et al. propose a new attribution method based on information bottlenecks. Neither of these works formalize the implicit validity assumptions behind fidelity metrics, nor do they provide tools to test whether these assumptions hold for a given model–perturbation pair. Our contribution is orthogonal: we identify the two fundamental assumptions (P1, P2) required for fidelity metrics to behave meaningfully and provide lightweight diagnostic measures (DROP, PSim) that test these assumptions directly. This shifts the discussion from improving attribution or evaluation metrics to validating whether the metrics themselves rest on valid behavioral premises.
>
> **W3:** "The study involves tens of millions of forward passes but....."
> **Rebuttal:** While it is intuitive that fidelity metrics depend on the perturbation used, prior work has not identified why they behave inconsistently or which specific assumption failures cause these inconsistencies. Our study goes beyond confirming intuition: it provides (i) a formal decomposition of the validity assumptions underlying fidelity metrics, and (ii) diagnostic measures (DROP and PSim) that quantify precisely where and how these assumptions fail. The large-scale experiments (with millions of forward passes) are necessary to show that these assumption violations are systematic across models, datasets, and perturbations, thereby offering an explanatory mechanism rather than reiterating a known empirical observation.
>
> **W4:** "The work is restricted to image-based saliency metrics, whereas ..... "
> **Rebuttal:** We study images because, unlike tabular or time-series data where features can be removed directly as spatial coherence is not relevant, image “deletion” necessarily involves modifying pixels, often inducing OOD behavior. This makes fidelity-metric assumptions far more fragile in vision, which is precisely the setting where our diagnostic framework is needed.
>
> **Q1:** "Why is Gaussian blur the most consistent perturbation type?...."
> **Rebuttal:** Gaussian blur performs better empirically, but we do not claim it as the optimal perturbation. We recommened that determining principled criteria for selecting perturbations is an important future direction in the Conclusion (Section 6) of the submitted paper.
>
> **Q2:** "How do DROP/PSim scores correlate with human-annotated explanation quality, if at all?"
> **Rebuttal:** Human-annotated “ground truth” regions do not necessarily reflect where a model actually attends or relies on, and several studies [1],[2],[3] caution against using human labels as a proxy for model reasoning. Since DROP and PSim are designed to test metric-validity assumptions rather than explanation quality, correlating them with human annotations would not be meaningful and could even be misleading. Our goal is to diagnose whether fidelity metrics behave consistently under perturbations, independent of whether humans would mark similar regions.
> [1] Colin et. al., “What I Cannot Predict, I Do Not Understand: A Human-Centered Evaluation Framework for Explainability Methods”, NeurIPS 2022
> [2] Geirhos  et. al., “Imagenet-Trained CNNs Are Biased Towards Texture; Increasing Shape Bias Improves Accuracy And Robustness”, ICLR 2019
> [3] Ebrahimpour et. al., “Do Humans Look Where Deep Convolutional Neural Networks “Attend”?”, 2019

---

> ### Author Response · Authors · 2025-12-04
> **Authors' rebuttal to Reviewer rfHB (part 2)**
>
> **Q3:** "Would the same inconsistency patterns appear in non-visual modalities........"
> **Rebuttal:** The inconsistency we analyze arises precisely because “deletion” is ill-defined in images: a pixel cannot be removed, only modified, and even small modifications typically push the sample off the natural-image manifold. This makes perturbation-based fidelity metrics particularly fragile in vision. In contrast, non-visual modalities such as tabular or text data allow clean feature-removal operations (e.g., dropping a column), and therefore do not suffer from the same ambiguity. Our study focuses on the domain where the problem is most severe and most widely observed.
>
> **Q4:** "Given the massive computational load, can DROP/PSim be approximated efficiently....."
> **Rebuttal:** We do not iterate over all pixels for computing DROP and PSim but use a small random subset (50 pixels/segments), which keeps the computational cost tractable even for large-scale experiments. The subset-ranking preservation result in the supplementary material ensures that this sampling preserves the relevant ordering properties. Hence, the reported results are already based on an efficient approximation rather than full-pixel evaluation.

---

### Official Review · Reviewer_e3wH · 2025-10-29

**Soundness:** 2
**Presentation:** 3
**Contribution:** 2
**Rating:** 2
**Confidence:** 4

**Summary:**

Common "fidelity" metrics in explainable AI for saliency maps are inconsistent, i.e., the metrics differ based on perturbation type.
The paper proposes two simple metrics to measure properties of a model: roughly, whether the probabilities drop under perturbation (a type of monotonicity property) and whether the pixel rankings are consistent across perturbation types.
DROP metric merely determines if the probability goes down or not.
PSIM metric simply averages pixel importance over multiple perturbation types.
The paper than examines multiple perturbation types across a range of different models and concludes that most models do not satisfy the proposed properties.

**Strengths:**

- The paper poses an interesting question: Are explanation methods based on Pixel Importance Rankings actually a good measure of fidelty?
- The metrics are simple to compute.

**Weaknesses:**

- The paper does not justify that [P1] and [P2] are required. While these seem relatively intuitive, it is not clear why these are required. Is this true just based on the definition of Pixel Importance Rank (PIR) and where is the original definition of PIR? Does the definition of PIR assume these are these "implicit" assumptions that the authors feel are appropriate? If it is not in the definition of PIR, why must these be true? If it is in the definition, please provide the best justification for them from the original paper(s).
  - Regarding PSim, it is claimed that these should have the same ranks IF "the model is consistent". What is meant by a model being consistent here? Is this formally proved? I'm not sure that a model that fails this test is inherently "inconsistent", it would just have different rankings based on different perturbation models. Again, not sure this is required to evaluate model explanations and/or the model.

- Overall, the story of the paper seems incremental. Other prior works have observed that perturbation type is important when defining PIR metrics. Also, it is unsurprising that the rankings of pixel are not the same using different perturbation types. What is perhaps a more interesting question is whether the explanation method's ranking differs significantly depending on the perturbation type. However, the paper does not answer this more important practical question.

- The paper lacks any significant theoretical or methodological advances. It proposes some simple measures for monotonicity and differences across perturbations but there is nothing particularly insightful about these metrics and the basis for [P1] and [P2] is not well justified. It is unclear if those points are actually required for the metrics to be reasonable.

- (Formatting) It looks like you used "\epsilon" instead of "\in" for summation subscripts, e.g., Eqn 6 $\phi \epsilon \mathcal{N}$.

**Questions:**

- How does this relate to [Wang & Wang, 2024]? In this paper, they explicitly increase the probability of the class in a deletion metric. As in, the probability doesn't go down. The perturbation is deletion and it is cumulative I think.

[Wang & Wang, 2024] Wang, Y. &amp; Wang, X.. (2024). Benchmarking Deletion Metrics with the Principled Explanations. <i>Proceedings of the 41st International Conference on Machine Learning</i>, in <i>Proceedings of Machine Learning Research</i> 235:51569-51595 Available from https://proceedings.mlr.press/v235/wang24br.html.

- How do you justify the set of perturbation types in DROP? This seems somewhat arbitrary. Is there a theoretically best version of this?

---

> ### Author Response · Authors · 2025-12-03
> **Authors' rebuttal to Reviewer e3wH (part 1)**
>
> We thank the reviewer for the comments. Our comments regarding the weaknesses, Questions posted by reviewer are as below:
>
> **W1:** "The paper does not justify that [P1] and [P2] are required. ...."
> **Rebuttal:** We respectfully clarify that [P1] and [P2] are not assumptions about how Deep Learning models should behave in general, nor are they behavioral claims we impose on the models. Instead, they are the implicit assumptions required by perturbation-based fidelity metrics themselves. These metrics only function as intended if a model satisfies (i) monotonic probability drop when relevant evidence is perturbed (P1), and (ii) proportionality between probability drop and pixel relevance (P2). This is standard across fidelity metrics such as AOPC, Deletion AUC and Faithfulness etc., where these behaviors are relied upon but never explicitly verified.
> Section 2.1 of the submitted paper formalizes these assumptions and shows why they are necessary for deletion-based fidelity metrics to be meaningful. This rationale is consistent with prior analyses such as FRIES (Bora et al., 2024), which similarly identifies these implicit dependencies when modeling the failure modes of fidelity metrics.
> Regarding “consistency,” Section 2.1 formalizes this precisely: under the assumptions required by fidelity metrics, pixel rankings produced under different perturbations should be approximately invariant (Eq. 5). PSim is therefore not an arbitrary similarity measure but the operational test of whether Assumption (P2) holds. **A low PSim score does not indicate that a model can be labelled as “inconsistent” but that it is inconsistent with the requirements of the fidelity metrics**, which is the central question of the paper.
>
> **W2:** "Overall, the story of the paper seems incremental......"
> **Rebuttal:** We respectfully disagree with the assessment that the contribution is incremental. Prior work such as Tomsett et al. (discussed in Section 1.1) has reported that fidelity metrics behave inconsistently across perturbation types, but it does not explain why this inconsistency arises. FRIES (discussed in Section 1.1) goes one step further by estimating inconsistency through supervised models that learn to predict metric disagreement, but it does not provide a diagnostic explanation of the root cause either and requires training additional models.
> Our paper addresses a different and more fundamental question. We analyze the underlying assumptions that perturbation-based fidelity metrics implicitly rely on, namely monotonicity (P1) and proportionality (P2), and we show that these assumptions systematically fail across common model–perturbation settings. DROP and PSim are designed as lightweight conformity tests for these assumptions. They do not require training any new models and directly reveal the structural reasons why fidelity scores disagree.
> We respectfully clarify that our proposed PSim metric measures exactly what the reviewer raises as a more interesting practical question: “whether the explanation method's ranking differs significantly depending on the perturbation type”. PSim quantifies the degree of rank invariance across perturbations through an RBO-based similarity measure. If rankings were stable, PSim would approach one. Our extensive results show that PSim is often low, indicating substantial ranking disagreement. Therefore, the paper directly answers the reviewer’s question in a principled and formalized manner.

---

> ### Author Response · Authors · 2025-12-03
> **Authors' rebuttal to Reviewer e3wH (part 2)**
>
> **W3:** “The paper lacks any significant theoretical or methodological advances……”
> **Rebuttal:** We respectfully disagree with the claim that the paper lacks theoretical justification for P1 and P2. These are not assumptions we introduce arbitrarily; they are the validity conditions implicitly required by perturbation-based fidelity metrics themselves. Section 2 of the submitted paper formally derives how fidelity metrics such as AOPC, Deletion AUC, and Faithfulness etc., rely on (i) monotonic decreases in class probability when relevant evidence is perturbed (P1) and (ii) proportional decreases aligned with pixel relevance (P2).
> This interpretation is consistent with prior work: FRIES (as discussed in Section 2) also identifies P1 and P2 as structural assumptions implicitly encoded in the way fidelity metrics function. FRIES treats violations of these assumptions as the root cause of inconsistency and builds supervised models to estimate this inconsistency, whereas our work provides a lightweight diagnostic that does not require training new models.
> DROP and PSim are therefore not “simple differences across perturbations” but are the operational tests that correspond directly to the theoretically derived conditions in Section 2. They evaluate whether a model-perturbation pair satisfies the assumptions that fidelity metrics require in order to produce meaningful scores. If P1 and P2 fail, then fidelity metrics become unreliable regardless of the explanation method. This represents a theoretical advance over prior observations, because it explains the mechanism behind the inconsistency rather than merely reporting it.
>
> **W4:** " (Formatting) It looks like you used "\epsilon" "
> **Rebuttal:** We thank the reviewer for pointing out the formatting issues. We have fixed it in the revised version.
>
> **Q1:** "How does this relate to [Wang & Wang, 2024]? In this paper, they explicitly....."
> **Rebuttal:** Wang and Wang (2024) is based on a fundamentally different premise. Their work optimizes the ordering of feature deletions that maximizes a deletion metric and treats this optimal ordering as the “principled explanation” for the model. This approach implicitly assumes that the deletion metric itself defines the ground-truth notion of relevance, which is a strong assumption and does not hold in general, since deletion metrics are known to be sensitive to perturbation choice and can induce severe out-of-distribution effects. Our does not make such assumptions. Instead, we analyze the assumptions that deletion-based fidelity metrics implicitly require (P1 and P2) and show that real models often violate them, which explains the inconsistency across perturbation types. Therefore, the two works address different questions: The reference provided by the reviewer analyzes the optimal behavior of the metric, whereas we analyze whether the fidelity metric’s assumptions hold for the model or not.
>
> **Q2:** “How do you justify the set of perturbation types in DROP? This seems somewhat arbitrary. Is there a theoretically best version of this?”
> **Rebuttal:** The perturbation set used in DROP directly follows established practice in the fidelity-metric literature. Prior works that analyze fidelity behavior including Tomsett et al. (AAAI 2020) and FRIES (Bora et al., Pattern Recognition 2026) evaluate perturbation-based metrics across multiple widely used perturbation operators, because different masking functions induce different model behaviors. Our goal is identical: DROP tests whether assumption P1 holds under the same perturbations that fidelity metrics themselves employ. The choice is therefore not arbitrary but aligned with standard perturbation families used in fidelity metrics in prior works (i.e. zero, mean, noise, blur, inpainting, min/max). Since fidelity metrics do not define a theoretically “best” perturbation, evaluating conformity across this established perturbation set is necessary to reflect actual usage. We added citation to these papers in the Section 4.1 where we provide details of the perturbations used in our experiments.
> A theoretically “best” perturbation is difficult to define because fidelity metrics are inherently operator-dependent, and even stable pertubations like Gaussian blur introduce some distributional shift. Our results show that perturbations should not be chosen arbitrarily, and we mentioned this in the Conclusion (Section 6) that future work should work to identify principled criteria for selecting appropriate perturbations is an important direction that our conformity analysis motivates.

---

### Official Review · Reviewer_no2c · 2025-11-07

**Soundness:** 2
**Presentation:** 3
**Contribution:** 2
**Rating:** 4
**Confidence:** 4

**Summary:**

This paper investigates the underlying reasons for the widely reported inconsistency of perturbation-based fidelity metrics in XAI. The authors posit that these inconsistencies stem from the violation of two fundamental assumptions inherent to such aspects: “[P1] There is a drop in the output probability when a pixel is perturbed; [P2] The magnitude of drop in output probability is proportional to the relevance of the pixel”. To formalize and test this hypothesis, the paper introduces two novel conformity measures: DROP and PSim. The authors conduct a large-scale empirical study involving five DL three datasets, nine distinct perturbation types, and two perturbation schemes (pixel-wise and segment-wise). The results demonstrate that both assumptions are frequently and significantly violated across most model-perturbation settings. And this is why perturbation-based fidelity metrics are inconsistent.

--soundness--
The paper posits that inconsistent fidelity metrics may stem from violations of assumptions [P1] and [P2] However, the proposed PSim metric and the experimental design are insufficient to substantiate these claims.
--contribution--
•	Importance of the Question: The reliability of evaluation metrics is a critical, foundational issue in XAI. Saliency methods are widely used, but the community lacks consensus on how to evaluate them.
•	Originality: The key originality lies in the conceptual reframing of the problem. Instead of treating inconsistency as a property of the metric, the authors identify it as a failure at the more fundamental model-perturbation interaction level.

**Strengths:**

•	Conceptual Insight and Originality: The paper's primary strength is its novel conceptualization of the problem. By shifting the focus from the fidelity metrics themselves to the underlying assumptions about model behavior, it provides a deeper and more fundamental explanation for the observed inconsistencies. This is a significant step forward from prior work.
•	Clarity and Presentation: The paper is exceptionally clear in its writing, structure, and presentation of results. The core ideas are communicated effectively, making the work accessible and its implications easy to grasp.

**Weaknesses:**

•	Random sampling: The practical implementation of the random sampling methodology introduces significant vulnerabilities that challenge the validity of the experimental conclusions. Although the underlying premise—that relative importance rankings are preserved in a subset—is correct, the approach is undermined by two fundamental limitations. First, the chosen sample size (e.g., 50 pixels out of 299 ×299, 224 ×224 or 600 ×600 pixels) is exceptionally small, raising concerns about the statistical power and significance of the findings. Second, and more critically, the method fails to address the problem of sample representativeness. An unbiased random draw from a typical image will, with high probability, yield a sample dominated by low-importance pixels with little variance in their contribution to the model's decision. This lack of diversity renders the rank-based evaluation metric unstable and incapable of meaningfully differentiating between the performance of various explanation techniques. Therefore, the reliability of the results becomes contingent on the fortuitous acquisition of a well-distributed sample, and a typical, uninformative sample would render the subsequent analysis and comparisons empirically weak.

•	The PSim score is a relative measure; it only reveals how similar the pixel rankings of two perturbation are to each other. A high PSim score between two methods does not prove that [P2] The magnitude of drop in output probability is proportional to the relevance of the pixel.

•	The experiment results may be caused by out-of-distribution data.

**Questions:**

•	Why conducting the analysis of adversarially trained models?
•	See Weaknesses

---

> ### Author Response · Authors · 2025-12-03
> **Authors' rebuttal to Reviewer no2c**
>
> We thank the reviewer for the comments. Our comments regarding the weaknesses, Questions posted by reviewer are as below:
>
> **W1:** "Random sampling: ...."
> **Rebuttal:** Prior works like FRIES (Bora et al., PR 2026), Sanity Checks for Saliency Metrics (Tomsett et al., AAAI 2020), as discussed in Section S2, evaluate model behavior using randomly selected pixels or regions, rather than exhaustively perturbing all pixels. Our sampling procedure is consistent with this established practice of the prior works.
> Additionally, although a subset of sampled pixels may appear to have low and similar relevance when discretized, they are not identical. If all sampled pixels indeed had the same low importance, then:
> •	DROP values would be identical across perturbation types, and
> •	PSim = 1 after sorting, because the relative order would be identical.
> However, this does not occur in our experiments: the differences in DROP and PSim scores across perturbation types empirically demonstrate that the sampled pixels carry subtle but meaningful differences in influence, which fidelity metrics react to. These differences are precisely what fidelity metrics depend on and what our conformity measures are designed to diagnose.
> Additionally, when computing PSim, we preserve the relative order of the sampled pixels, if their importances are same. If all sampled pixels had equal influence, the rank order would collapse into ties and PSim would converge to 1 for all perturbations. The fact that PSim < 1 and varies across perturbation types demonstrates that the sampled subsets do contain discriminative signal, even in low-relevance regions.
>
> **W2:** "The PSim score is a relative measure; it only reveals how...."
> **Rebuttal:** We wish to clarify that the theoretical basis for PSim is already established in the main paper. Section 2.1 formalizes Assumption [P2] through Equation (5), which requires the Rank Biased Overlap (RBO) between the pixel importance rankings under perturbations ϕ and ψ to be ≈1. Section 2.3 then defines PSim explicitly as the operational conformity measure corresponding to this assumption, i.e., PSim quantifies the deviation from the requirement expressed in Equation (5). Further, the subset-ranking preservation result (proved in Supplement Section S2) guarantees that pairwise ranking relations are maintained even when computed on sampled pixels. Therefore, PSim is not intended to “prove [P2]”; rather, it is the formal measure that measures the extent to which [P2] is violated. The theoretical linkage is provided in the mentioned sections of the main (2.1 and 2.3) and the supplementary (S2) parts of the submitted paper.
>
> **W3:** "The experiment results may be caused by out-of-distribution data."
> **Rebuttal:** We agree that certain perturbations can push images outside the natural data manifold. Since existing fidelity metrics operate under this same perturbation protocol and do not correct for such OOD drift, we follow the standard setup to remain consistent with their assumptions. Thus, OOD effects are not confounding the results rather they are the mechanism that fidelity metrics fail to address, and DROP/PSim highlight and quantify this failure in a principled way.
>
> **Q1:** "Why conducting the analysis of adversarially trained models? "
> **Rebuttal:** We include adversarially trained models to test whether increased model robustness improves conformity to the assumptions underlying fidelity metrics. Adversarial training is widely known to make models more stable under perturbations, so it provides a **natural stress test** for [P1] and [P2]. Our results show that even these more robust models do not satisfy the required conformity conditions, which strengthens the generality of our findings. This analysis does not change the core experimental setup; it only evaluates whether robustness affects conformity.

---

### Official Review · Reviewer_S6of · 2025-11-11

**Soundness:** 3
**Presentation:** 3
**Contribution:** 3
**Rating:** 4
**Confidence:** 2

**Summary:**

The paper addresses a critical issue in Explainable AI (XAI) - saliency maps (like CAM) are widely used to interpret deep learning models, but assessing their quality through fidelity metrics is unreliable. Different fidelity metrics behave inconsistently under different perturbations, making it unclear which explanations are actually trustworthy.

**Strengths:**

1. The introduction of conformity measures (DROP and PSim) that directly test metric assumptions is genuinely novel. This shifts from empirical observation to theoretical diagnosis.

2. Distinctive from FRIES: While using similar "foundational primitives," the authors cleverly repurpose them for diagnostic rather than predictive purposes; a supervised model is needed. This is an elegant conceptual distinction.

**Weaknesses:**

Unclear decision boundaries: Tables 1 and 2 show DROP and PSim scores with means and standard deviations, but there's no clear guidance on what constitutes acceptable conformity. For example, is DROP=0.504±0.131 good or bad? When should practitioners be concerned?

KDE-based cutoffs are ad-hoc: The paper mentions using KDE with cutoffs at 0.80, 0.85, 0.90, and 0.95, but the rationale for these specific thresholds is unclear. Why these values? How sensitive are conclusions to threshold choice?

Probabilistic interpretation is vague: Lines 358-365 mention that "estimated probabilities for DROP scores to be above the cutoffs...were low, and the variants of Gaussian Blur showed relatively higher probabilities than other perturbations." This is descriptive but doesn't translate to actionable guidance: Should I avoid fidelity metrics entirely? Switch perturbations? Use different metrics?

**Questions:**

like weakness

---

> ### Author Response · Authors · 2025-12-03
> **Author Rebuttal to Reviewer S6of**
>
> We sincerely thank the reviewer for the thoughtful and positive assessment of the paper’s conceptual novelty and its distinction from FRIES (Bora et. al. 2025). We address the weaknesses below.
>
> **W1:** "Unclear decision boundaries: Tables 1 and 2 show DROP and PSim scores with means and standard deviations..."
> **Rebuttal:** DROP and PSim are already defined in the paper as absolute conformity measures in [0,1], where 1 represents full satisfaction of the assumptions required by fidelity metrics (P1/P2) and any value below 1 indicates a violation. This interpretation is explained in Sections 5.1 and 5.3 and used consistently in experiments. For example, a DROP score of 0.50±0.13 for a perturbation indicates that the condition P1 holds only about half the time under that perturbation, which signals low conformity.
>
> Because full conformity corresponds to a score of 1, there is no separate “decision boundary’’ beyond this ideal point. The scores are therefore directly interpretable, and practitioners can read any deviation from 1 as the extent to which a perturbation violates the fidelity metric’s required assumptions.
>
> **W2:** "KDE-based cutoffs are ad-hoc...."
> **Rebuttal:** The KDE-based cutoffs are not decision thresholds. As described in Sections 5.2 and 5.3 (and Supplement S2–S6), the values 0.80, 0.85, 0.90, and 0.95 are simply proximity bands used to quantify how much probability mass lies near the ideal value 1. They are not used to judge whether a perturbation is “acceptable,” but only to measure how close the DROP/PSim distributions are to full conformity. Any reasonable set of high-end thresholds (e.g., within 0.8–1.0) yields the same qualitative conclusion: the probability of scores approaching 1 is consistently low. Since the cutoffs serve only as descriptive proximity bands rather than decision thresholds, the usual notion of threshold sensitivity is not applicable in this context.
>
> **W3:** "Probabilistic interpretation is vague: Lines 358-365 mention that "estimated probabilities for DROP scores to be above the cutoffs....."
> **Rebuttal:** We respectfully clarify that our probabilistic analysis is not intended to prescribe which fidelity metric or perturbation a practitioner “should” use, but rather to quantify whether the assumptions that make those metrics meaningful are satisfied. In the Conclusions (Section 6) of our submitted paper, we have provided concrete, actionable guidance:
> (i) fidelity metrics should not be applied without first checking conformity through DROP/PSim;
> (ii) the exact perturbation with hyper-parameters must be reported, as different perturbations induce different conformity profiles; and
> (iii) perturbations should be theoretically justified rather than arbitrary masks.
>
> Our KDE-based tail probabilities are used only to show how infrequently conformity approaches the ideal value 1; this is not a prescription mechanism but a diagnostic tool. We also explicitly note that Gaussian blur showed comparatively higher conformity, while emphasising that selecting optimal perturbations is an open direction for future work.

---

### Author Response · Authors · 2025-12-03
**Overall Comment by Authors**

We thank all reviewers for their insigntful comments. We wish to clarify that our paper does not assume that Deep Neural Networks inherently satisfy the following properties (as explained in Sections 1 and 2 of the submitted paper):

P1: There is a drop in the output probability when a pixel is perturbed.

P2: The magnitude of drop in output probability is proportional to the relevance of the pixel.

On the contrary, P1 and P2 are the implicit assumptions made by fidelity metrics (discussed in Sections 1 and 2 of the submitted paper). **Our conformity measures (DROP and PSim), therefore, evaluate whether a model satisfies the preconditions on which these metrics rely, not whether the model “should” behave in a particular way**. The reviewers’ concerns broadly interpret P1 and P2 as model-behavior assumptions; in fact, they are validity requirements for fidelity metrics, and our empirical findings show that these requirements are violated in practice, leading to inconsistency in fidelity metrics.

---

### Meta-Review · Area_Chair_YNHS · 2026-01-04

**Summary:**

The paper examines why perturbation-based fidelity metrics for saliency maps yield different results depending on the perturbation. The authors argue that this happens because the implicit assumptions underlying these metrics often fail, and they propose DROP and PSim as quick checks of those assumptions across models/datasets/perturbations. Reviewers agree the topic is important and the experiments are thorough. Still, there was a limited push to accept, mainly due to novelty concerns and skepticism that assumption + diagnostics add enough beyond what is already known about perturbation dependence.

**Reviewer Concerns:**

issues partially addressed:
* clarification about claims
* clearer positioning relative to prior work
* explanation of the role of adversarially trained models as stress tests rather than core contributions
* responses to presentation and notation issues

issues that remain concerns:
* several reviewers question whether assuptions are truly necessary requirements for fidelity metrics, or whether they reflect one possible (but not universal) interpretation of faithfulness
* novelty concerns: multiple reviewers view the contribution as incremental, formalizing intuitions already present in prior work
* DROP and PSim are seen primarily as diagnostic or descriptive tools, with limited new theoretical insight into why specific perturbations or model properties cause failures
* methodological concerns about random sampling of pixels/regions and representativeness, which the rebuttal defends by precedent rather than fully resolving
* limited scope to vision models and CNNs, leaving open questions about generality to other architectures or modalities
* lack of strong actionable guidance for practitioners beyond “be careful” and “report perturbations.”

**Reviewer Scores:**

* S6of: 4, likely unchanged
* no2c: 4, likely unchanged
* rfHB: 4, likely unchanged
* e3wH: 2, unchanged
* XEKX: 2, unchanged

---

### Decision · Program_Chairs · 2026-01-26

Reject